# The mechanism of cell-cycle-dependent proteasomal degradation of archaeal ESCRT-III homolog CdvB in *Sulfolobus*

Yin-Wei Kuo [ID][1,2], Jovan Traparić [ID][1,2], Sherman Foo [ID][1] & Buzz Baum [ID][1✉]

## Abstract

Protein degradation orders events in the cell division cycle in eukaryotes, bacteria, and archaea. In eukaryotes, chromosome segregation and mitotic exit are triggered by proteasome-dependent degradation of securin and cyclin B, respectively. Recent findings show that the archaeal proteasome also targets substrates, including CdvB, for degradation in a cell-cycle-dependent manner in *Sulfolobus acidocaldarius*—an experimentally tractable archaeal relative of eukaryotes. Here, using CdvB as a model substrate to explore the mechanism of cyclic protein degradation, we demonstrate that the C-terminal broken-winged helix of CdvB, previously shown to bind CdvA, is sufficient to render a fusion protein unstable as cells progress through division. We show that the rate of CdvB degradation accelerates during division in part due to a cell-cycle-dependent increase in expression of the proteasome-activating nucleotidase (PAN), under the control of a cyclically expressed novel transcription factor "CCTF1" that represses PAN expression. Taken together, these findings reveal mechanisms by which archaea, despite lacking cyclin-dependent kinases, control proteasome-mediated degradation to order events during cell division.

**Keywords** Cell Division Cycle; Archaea; Proteasome; ESCRT-III
**Subject Categories** Cell Cycle; Evolution & Ecology; Post-translational Modifications & Proteolysis

## Introduction

Orderly progression through a round of the cell division cycle requires certain proteins to be expressed and degraded at specific times (Fry and Yamano, 2006; Fatima et al, 2022; Tarrason Risa et al, 2020). In eukaryotes, a wave of protein degradation at mitotic exit resets the cell cycle and triggers chromosome segregation, cytokinesis, and replication origin licensing. This process is initiated upon satisfaction of the spindle assembly checkpoint via the recruitment of Cdc20 to the CDK-activated Anaphase Promoting Complex (APC/C) (Musacchio, 2015). The Cdc20-APC/C complex then ubiquitylates proteins that contain a destruction-box motif (RXXLXXXXN), including Cyclin A, Cyclin B, and Securin, targeting them for proteasome-mediated degradation (Glotzer et al, 1991; McAinsh and Kops, 2023). As cells exit mitosis and enter G1, Cdc20 is then swapped out for Cdh1, which targets additional substrates of the APC/C that typically contain a KENXXXN box motif, like Cdc20, for ubiquitylation and degradation (Pfleger and Kirschner, 2000; Davey and Morgan, 2016). This extended wave of cyclic proteasome-mediated protein degradation is therefore achieved by: (i) regulation at the level of a substrate's affinity for the degradation targeting machinery ensuring, for example, that Cyclin A is degraded before Cyclin B, and (ii) changes in the degradation machinery itself, with Cdh1-APC/C taking over from Cdc20-APC/C at late stages of mitotic exit.

Despite lacking eukaryotic-like cell cycle regulators such as cyclins, CDKs, or the APC/C, Thermoprotei like Sulfolobales have an ordered cell cycle that appears superficially similar to that of many eukaryotes (Cezanne et al, 2024; Lindås and Bernander, 2013). While it is not yet known how this is regulated, progression through the division phase in *Sulfolobus* is accompanied by a wave of proteasome-mediated protein degradation that resembles the one observed in eukaryotes exiting mitosis and undergoing cytokinesis. A key target of the *Sulfolobus* proteasome at division is CdvB (Tarrason Risa et al, 2020; Liu et al, 2025), an ESCRT-III homolog (Samson et al, 2008; Lindås et al, 2008). The timely degradation of CdvB is critical for orderly cell cycle progression, since CdvB must first accumulate to form a medial polymeric ESCRT-III ring before DNA segregation and cytokinesis can occur (Parham et al, 2025). Then, a few minutes later, the disassembly of the CdvB polymer and concomitant CdvB degradation are required for constriction of the cytokinetic CdvB1/B2 ring and abscission (Tarrason Risa et al, 2020; Harker-Kirschneck et al, 2022).

The sequence of events that leads to cell division in *Sulfolobus acidocaldarius* can be summarized as follows: As G2 cells prepare for division, they begin to express a wave of cell division proteins, which include CdvA, CdvB, CdvB1 and CdvB2, and Vps4. CdvA is the first to accumulate (Samson et al, 2011), enabling it to form a medial ring that defines the division plane (Samson et al, 2011; Lindås et al, 2008). CdvA then recruits CdvB, through the

[1]Medical Research Council Laboratory of Molecular Biology, Cambridge CB2 0QH, UK. [2]These authors contributed equally: Yin-Wei Kuo, Jovan Traparić.
✉E-mail: bbaum@mrc-lmb.cam.ac.uk

interaction of the CdvB broken winged-helix domain with the E3B region of CdvA (Samson et al, 2011). Subsequently, CdvB1 and CdvB2 are recruited to the ESCRT-III ring, leading to the formation of a precisely positioned composite division ring (Hurtig et al, 2023), which is a prerequisite for DNA segregation to occur (Parham et al, 2025). Once cells are ready to divide, the CdvB polymer is then disassembled by the AAA-ATPase Vps4, facilitating the constriction of the CdvB1/B2 cytokinetic ring (Harker-Kirschneck et al, 2022). Monomeric CdvB is then degraded by the proteasome in a process that requires the proteasome-activating nucleotidase (PAN) (Hurtig et al, 2023; Tarrason Risa et al, 2020)—an archaeal homolog of the 19S proteasome subunit (Maupin-Furlow, 2011; Forouzan et al, 2012).

Although the formation and the timely destabilization of the CdvB ring are both known to play critical roles during division in *S. acidocaldarius*, it is not yet clear how cells ensure that CdvB is targeted for proteasome-mediated degradation at precisely the right time. Here, in exploring the mechanism of cyclic CdvB regulation in *S. acidocaldarius*, we map the regions of CdvB that contribute to its cyclic instability. This analysis identifies poorly structured regions of the C-terminal tail of CdvB that destabilize the protein across the cycle. In addition, it demonstrates a role for the broken winged-helix domain of CdvB—a previously reported interaction site of CdvA (Samson et al, 2011)—in the transient stabilization of the protein during the pre-division phase of ring assembly. Finally, we demonstrate a role for transcription factor-mediated changes in the levels of the PAN that contributes to the increase in CdvB degradation as cells initiate division. Taken together, these data reveal how sequences in the tail of CdvB that target the protein for degradation combine with general changes in the levels of the protein degradation machinery to regulate cell cycle progression in *Sulfolobus*.

# Results

## The C-terminal region of CdvB contributes to its rapid degradation

To understand the molecular origin of the rapid proteasome-mediated degradation of CdvB as *Sulfolobus* cells switch from a phase of CdvB ring assembly to CdvB1/B2-dependent ring constriction, we began by using flow cytometry to compare the degradation kinetics of CdvB relative to its two homologs, CdvB1 and CdvB2 (Fig. 1A–F). This revealed, as previously described (Hurtig et al, 2023), that while the three proteins form a composite ring of homopolymers in cells preparing to divide, the entire cellular pool of CdvB is degraded before division is complete (Fig. 1B). By contrast, CdvB1 and CdvB2 are gradually degraded several minutes later as cells pass from G1 into S phase (Fig. 1C,D; Appendix Fig. S1). Interestingly, the G1/S degradation of CdvB1 and CdvB2 also depends on the proteasome since, like CdvB, the rate of their degradation in G1 could be reduced by the addition of bortezomib, a proteasome inhibitor, to the medium (Fig. 1E,F). These data indicate that different proteins are targeted to the proteasome at different times in the *Sulfolobus* cell cycle—as is the case for cell cycle machinery that is degraded by proteasome in eukaryotic cells (Morgan, 2007).

These substrate-specific differences in the timing of proteasome-dependent protein degradation in dividing *Sulfolobus* cells prompted us to compare the domain architecture of the three homologs (Fig. 1G). Inspection of the sequences showed that CdvB has a longer C-terminal domain that contains a MIM2 motif, a linker region, and an additional domain containing a broken-winged helix (Appendix Fig. S2; Fig. 1G), which is conserved across Sulfolobales (Makarova et al, 2024) and has been reported to interact with the non-ESCRT-III division protein, CdvA (Samson et al, 2011).

To test whether the C-terminal region of CdvB is sufficient to induce fast proteasome-mediated degradation during division, we fused the C-terminal region of CdvB to an N-terminally HA (hemagglutinin)-tagged LacS protein: a beta-galactosidase variant taken from *Saccharolobus solfataricus* (see Appendix Fig. S3 for details on the choice of LacS). Note that the α5 helix of ESCRT-III core domain (preprint: Drobnič et al, 2025) was included due to a partial overlap with the annotated MIM2 motif (Samson et al, 2011) to avoid truncation of MIM2 in our construct. We then expressed this fusion protein from a plasmid under the arabinose-inducible promoter and, following induction with 0.2% arabinose, used Western blotting to compare the expression of the HA-LacS-CdvB[C-term] fusion protein with LacS fused to the C-terminal tail of CdvB1 and CdvB2, or an HA-LacS control.

While the LacS and the LacS fusion proteins carrying CdvB1 or CdvB2 C-terminal regions accumulated to high levels in asynchronous populations of cells, the CdvB C-terminal tail fusion was expressed at much lower levels under the same conditions (Fig. 2A). This implies that the C-terminus of CdvB contains information that renders the fusion protein unstable. We confirmed that this was due to proteasome-mediated protein degradation by showing that the HA-LacS-CdvB[C-term] fusion protein rapidly accumulated following the addition of a proteasome inhibitor, bortezomib (Appendix Fig. S4).

Next, to assess the impact of the CdvB tail on cyclic protein degradation, we used flow cytometry to evaluate the levels and timing of HA-LacS-CdvB[C-term] protein accumulation relative to HA-LacS-CdvB1/B2[C-term] proteins. Consistent with the Western blotting analysis, this flow cytometric analysis revealed that the CdvB[C-term] fusion protein had the lowest levels of HA expression (Fig. 2B). We then separated the population into G1 and Division phases (D-phase) using CdvA and DNA content as a guide (Fig. 2C) to obtain information about the levels of each fusion protein as they progress through the cell cycle. Interestingly, this analysis revealed that the HA-LacS-CdvB[C-term] fusion protein is rapidly degraded during the transition from the D-phase (2N DNA content and high CdvA) into G1 phase (1N DNA content), resulting in an approximately fivefold decrease in protein (equivalent to a G1/D phase HA-signal ratio of ~20% (Fig. 2D)). By contrast, the G1/D ratio for LacS, or the CdvB1[C-term] or CdvB2[C-term] fusion proteins was closer to 50%—the value expected for a stable protein that is evenly partitioned between two daughter cells during division (Fig. 2D). These data show that the C-terminal region of CdvB contains sufficient information to confer cyclic proteasome-dependent protein degradation on a fusion protein.

For endogenous CdvB to be degraded prior to division, the protein must first be removed from the polymeric ESCRT-III ring by the activity of the AAA-ATPase Vps4 as cells switch from a ring assembly phase to a ring constriction phase (Hurtig et al, 2023; Liu et al, 2025). To determine whether a similar process might contribute to the degradation of the HA-LacS-CdvB[C-term] fusion

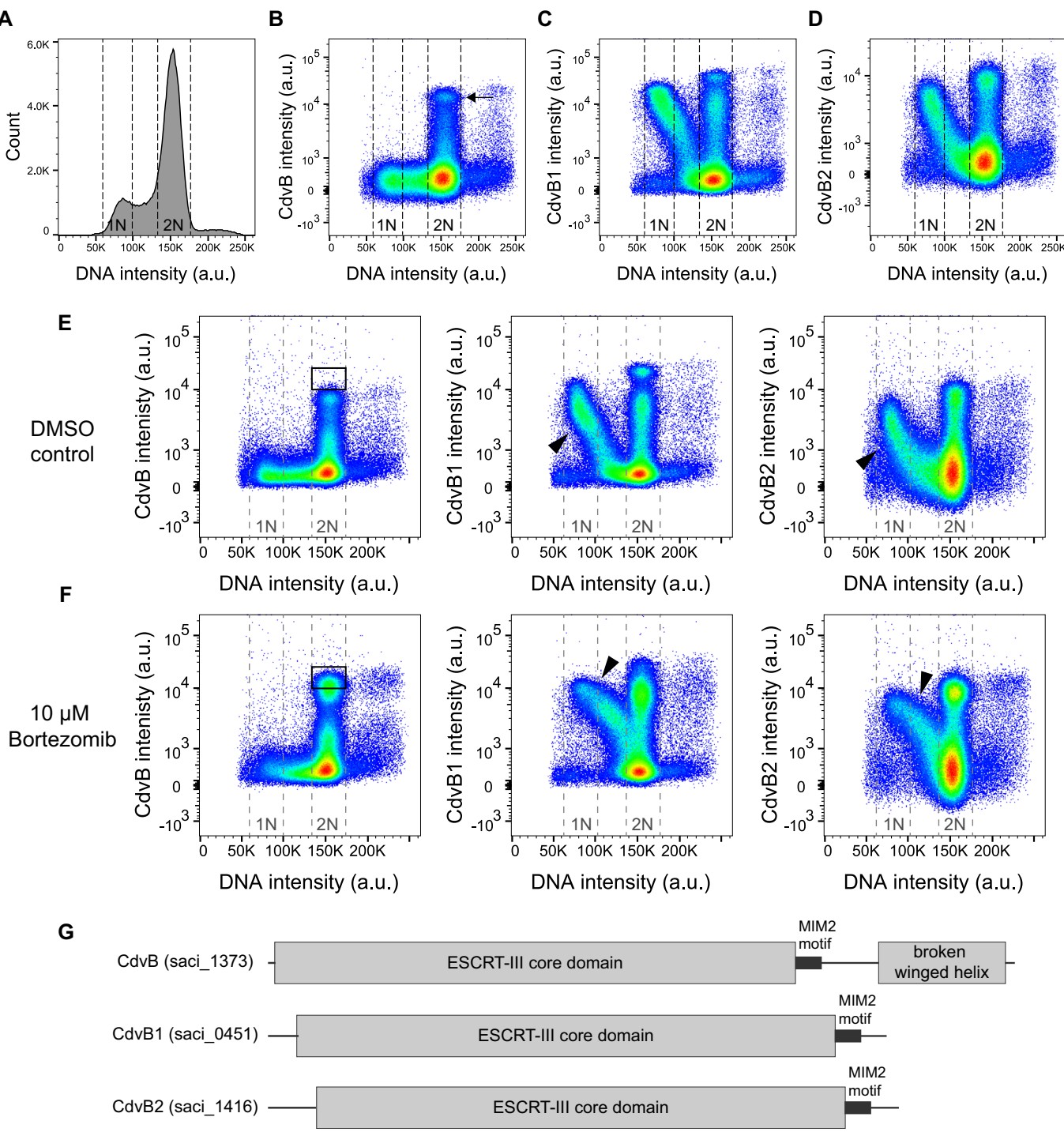

**Fig. 1. The ESCRT-III homologs of *Sulfolobus acidocaldarius*.**

(**A–D**) Representative flow cytometry analysis of asynchronous cultures of MW001 cells labeled for DNA (**A**), endogenous CdvB (**B**), CdvB1 (**C**), or CdvB2 (**D**) in a histogram and two-dimensional density plots. Cells in G1 and G2 phases of the cell cycle are indicated as 1N and 2N, respectively. Each dot represents a single event with the density gradient going from blue to red ($n = 2.5 \times 10^5$ events each). The population of cells with high levels of CdvB (arrow in (**B**)) is in the pre-constriction phase of division. (**E, F**) Flow cytometry plots of MW001 cells treated with (**E**) DMSO (control) or (**F**) 10 μM proteasomal inhibitor bortezomib, showing that proteasomal inhibition leads to an accumulation of CdvB (black boxes) and a reduction in the degradation rate of CdvB1 and CdvB2 (arrow heads indicate a decrease in the slope of G1-S phase degradation trajectories). (**G**) Schematic of the architecture of the three ESCRT-III homologs showing the conserved ESCRT-III core domain, the MIM2 motif, the connecting linker, as well as the broken winged-helix motif of CdvB. Source data are available online for this figure.

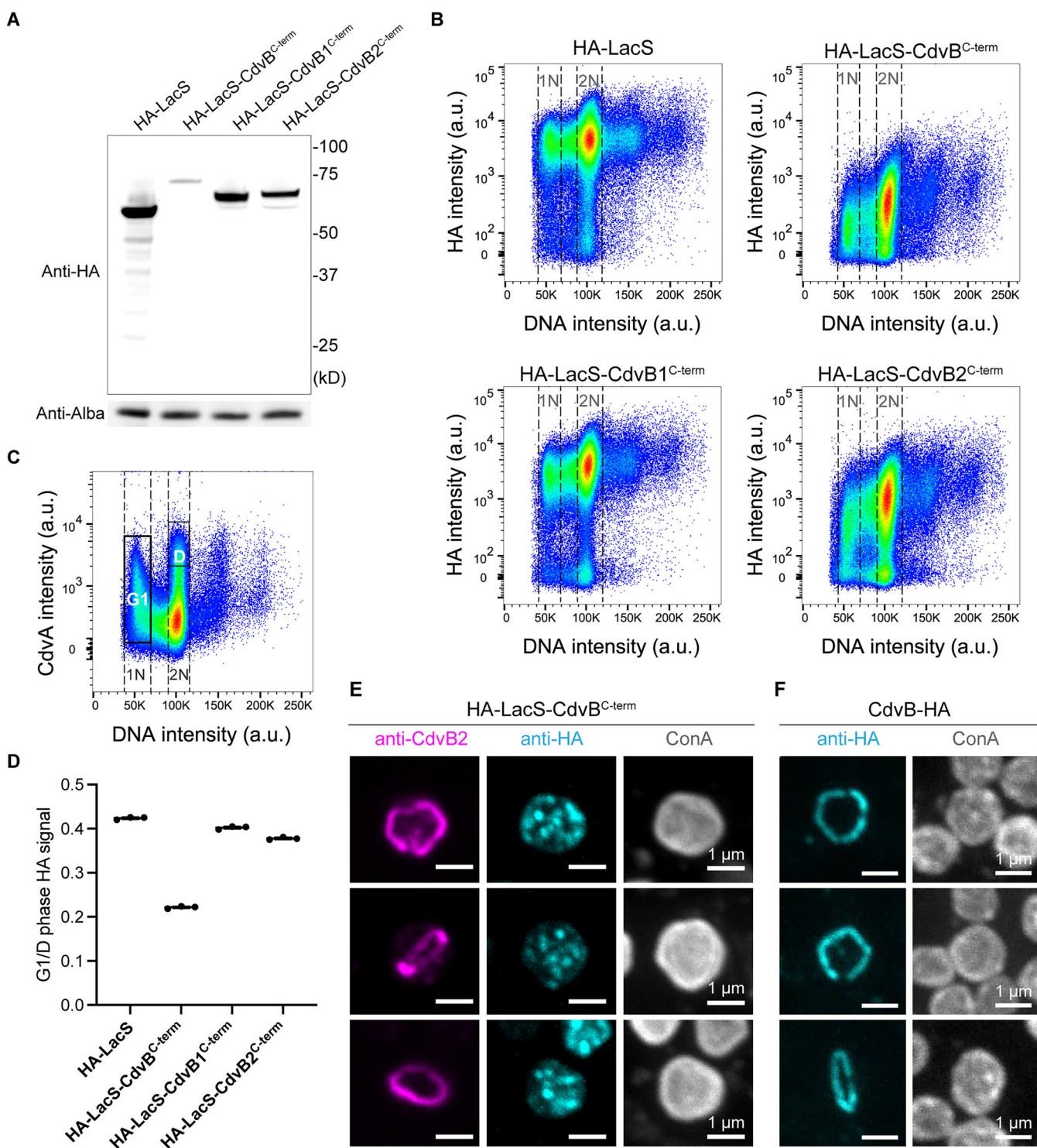

**Fig. 2. Effects of the C-terminal region of CdvB on LacS stability.**

(A) Representative Western blot of HA-tagged LacS fused with the C-terminal regions of the ESCRT-III proteins CdvB, CdvB1, or CdvB2 under the expression of the arabinose promoter. DNA-binding protein Alba was used as the loading control. (B) Flow cytometric analysis of HA-tagged LacS fused to the C-terminal regions of the ESCRT-III proteins CdvB, CdvB1, or CdvB2, plotted against DNA content. Cells in G1 and G2 phases of the cell cycle are indicated as 1N and 2N, respectively. Each dot represents a single event with the density gradient going from blue to red ($n = 2.5 \times 10^5$ events each). (C) Representative analysis of the CdvA levels in asynchronous *S. acidocaldarius* cells, showing the division phase (D-phase) and G1 phase. The division phase population typically corresponds to ~4% of total cells. (D) Quantification of the HA signal from (B) as a ratio of the average level in G1 and D-phases. (E, F) Example immunofluorescence images of HA-LacS-CdvB[C-term] (E) and full-length CdvB-HA (F) overexpressed under an arabinose-inducible promoter in cells in the pre-constriction stage. Source data are available online for this figure.

protein, we used fluorescence microscopy to test if the C-terminal tail of CdvB is sufficient to target LacS to the medial ESCRT-III ring in dividing *Sulfolobus* cells. This was not the case. HA-LacS-CdvB$^{C\text{-term}}$ appeared to be cytoplasmic and was not recruited to the division ring (Fig. 2E)—consistent with it lacking the ESCRT-III domain required for polymerization (preprint: Drobnič et al, 2025), while overexpressed full-length CdvB-HA was recruited to the division ring (Fig. 2F). These data indicate that the degradation of the fusion protein carrying the CdvB C-terminus is independent of ESCRT-III polymerization dynamics and must be regulated in some other way as cells initiate cell division.

## Dissecting the region of CdvB required for cyclic degradation

To map the specific signal(s) at the C-terminal region required for the cyclic degradation of LacS-CdvB$^{C\text{-term}}$, we generated a series of truncation constructs removing the α5 helix of the ESCRT-III core domain, the MIT-interacting motif (MIM2), and the broken winged helix domain in different combinations. We then ran a flow cytometric analysis (Fig. 3A–E; Appendix Fig. S5) to compare the levels of different fusion proteins in dividing cells (D-phase) and in G1 cells—using the G1/D-phase HA-signal ratio as a quantitative read-out of cyclic degradation.

As expected, the inclusion or removal of the α5 helix had little effect on the cyclic degradation of the C-terminal fusion construct since this is part of the ESCRT-III core domain (Δ160–183 and Δ184–261 in Appendix Fig. S5). On the other hand, deletion of the MIM2 motif (Δ183–193), enhances the cyclic degradation (Fig. 3C) by partially increasing the fusion protein stability in D-phase (Fig. 3D; Appendix Fig. S6), while still being efficiently degraded upon passage into G1 (Fig. 3E; Appendix Fig. S6). By contrast, the general instability that is characteristic of the CdvB tail was largely lost in the Δ194–261 deletion construct, which only retains the α5 helix and MIM2 domain (Fig. 3B–D), and in the 194–211 construct, which lacks everything but the linker that normally connects the MIM2 domain with the broken-winged helix (Appendix Fig. S5). Importantly, these two constructs (Δ194–261 and 194–211) lacking the broken winged helix domain showed little cyclic degradation (G1/D-phase HA ratio ~0.4; Fig. 3C; Appendix Fig. S5B). These data imply a role for the broken winged-helix domain of CdvB in cell cycle-dependent protein degradation.

In line with this conclusion, while a fusion protein that lacked the broken winged-helix domain but retained the MIM2 and linker regions (Δ212–253) still proved unstable (Fig. 3B,D,E), it failed to undergo increased degradation during the transition from D-phase to G1 phase (Fig. 3C). As a consequence, levels of the (Δ212–253) fusion protein were reduced by ~50% upon passage from D-phase into G1 (Fig. 3C), like the LacS and CdvB1 and CdvB2 fusion protein controls (Fig. 2D). Together, these data suggest there are at least two major destabilizing sequences in the C-terminal tail of CdvB. A region spanning the MIM2 motif and its flanking linker confers general instability on the fusion protein in a manner that is independent of the cell cycle (as demonstrated by the Δ212–253 construct). Second, the broken winged-helix domain targets a fusion protein for degradation as the cells transit from D-phase to G1 phase. In addition, a more in depth analysis of the data suggested that the broken-winged helix functions both to stabilize LacS during early division (the D-phase HA signal is higher in

Δ183–193, but lower in Δ212–253, Fig. 3D) and to destabilize the fusion protein at other stages of the cell cycle (the HA signal for the Δ183–193 construct is low in G1, but is higher for the Δ212–253 fusion, Fig. 3E). Since the broken winged-helix domain is expected to interact with CdvA (Samson et al, 2011), it is possible that the destabilization of CdvB as cells pass from D-phase into G1 in part reflects the loss of binding of the CdvB C-terminal tail to CdvA during the transition from the ring assembly to the ring constriction phase of division.

This analysis also revealed the Δ149–193 deletion construct that retains the broken winged-helix domain but lacks the neighboring α5-helix and MIM2 domains (Appendix Fig. S5A) accumulates to such high levels in the division phase that it blocks cell division (Appendix Fig. S5B). Since this mirrors the effects of expressing a CdvA construct that lacks the CdvB interaction site (Parham et al, 2025), which inhibits the transition from pre-division to constriction, it is likely that the broken winged-helix domain when expressed in high level interferes with the association of endogenous CdvB with CdvA during the ring assembly phase, leading to a division defect.

Taken together, these results suggest that the normal cyclic pattern of CdvB degradation is regulated by sequences within the MIM2 and linker region that render the protein unstable independent of the cell cycle stage, and by the broken winged-helix domain that preferentially stabilizes the protein during the ring assembly phase of division.

## The cyclic degradation of CdvB accords with the cyclic expression of proteasome-activating nucleotidase (PAN)

After identifying the broken-winged helix as part of the molecular signal that regulates cyclic degradation of CdvB, we next examined whether there are changes in the protein degradation machinery itself during the transition from ESCRT-III ring assembly to cytokinesis. We began by investigating the involvement of the proteasome-activating nucleotidase (PAN) in the degradation of CdvB, since previously published work had implicated it in the proteasome-mediated degradation of CdvB (Tarrason Risa et al, 2020). Consistent with this earlier work, overexpression of the ATP-hydrolysis-deficient mutant (E237Q) of PAN led to increased division failure as manifested by the accumulation of polyploid cells (Fig. 4A,B). We also observed a residual pool of CdvB protein in newly divided G1 cells expressing this dominant negative PAN, which was absent from G1 cells in the empty vector control strain (Fig. 4A black boxes, Fig. 4C) and from G1 cells in the MW001 background strain (Tarrason Risa et al, 2020). While we did not previously observe G1 cells with residual CdvB after treatment with high dosage of the proteasome inhibitor (Fig. 1E,F; Appendix Fig. S7A,C), which blocks cell division (Tarrason Risa et al, 2020), a population of G1 cells containing residual CdvB was observed when cells were exposed to low dosage of bortezomib (Appendix Fig. S7B), indicating that residual levels of CdvB can be observed in cells passing from division into G1 cells if degradation is partially compromised.

Since our previous work suggested an important role for CdvB degradation in the execution of cytokinesis (Tarrason Risa et al, 2020), which recently published work suggests is less pronounced in other species (Liu et al, 2025), to determine how the failure to degrade CdvB influences division in *S. acidocaldarius*, we fixed,

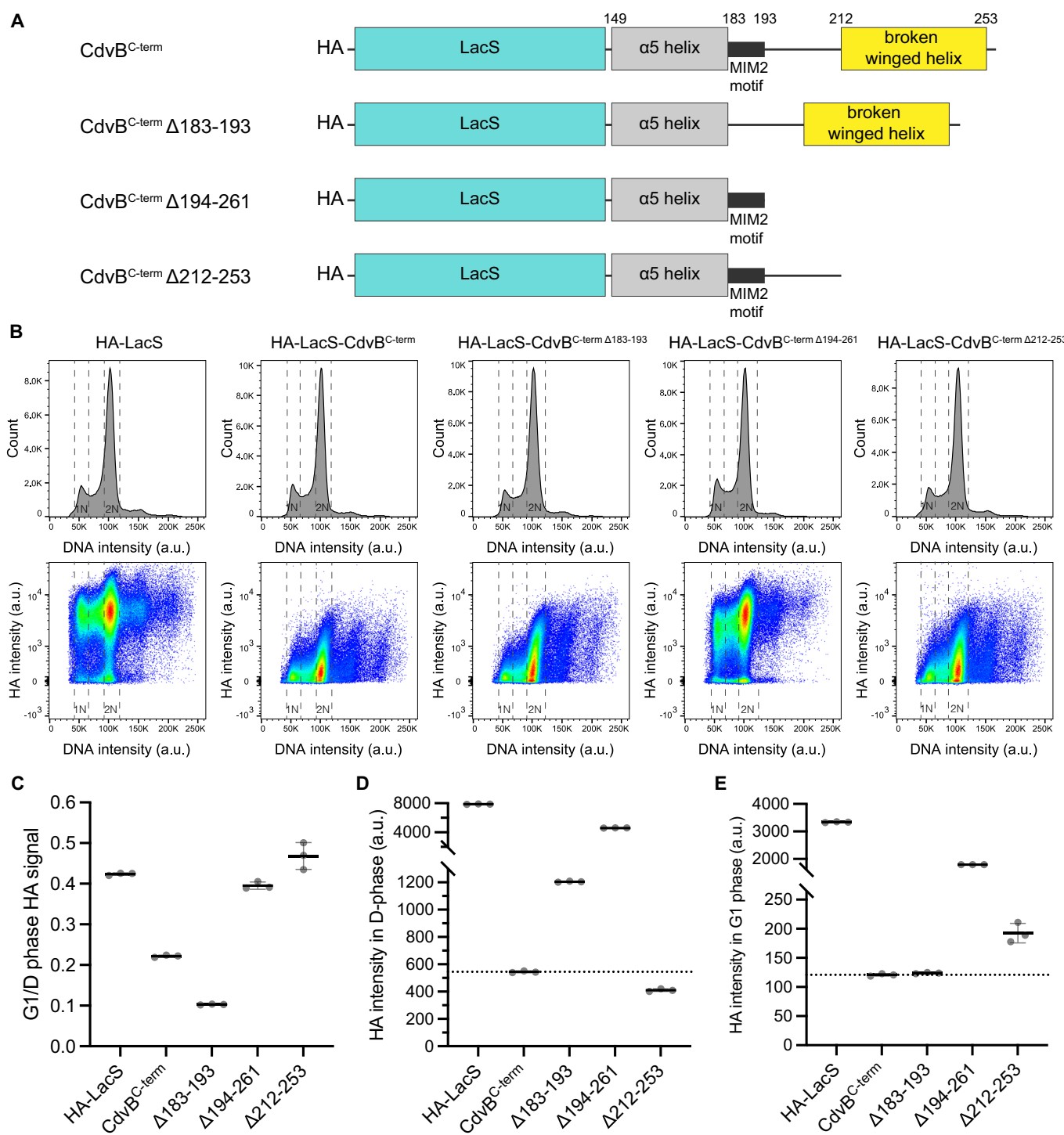

**Fig. 3. Identification of the region required for cyclic degradation at the C-terminal region of CdvB.**

(A) Schematic diagram of the LacS-CdvB[C-term] constructs used in this figure. (B) Representative flow cytometry histograms and scatter plots of HA-LacS fused with the CdvB[C-term] truncation constructs. (C) Quantification of cyclic degradation of indicated truncation constructs using the HA signal ratio in G1/D-phase. (D, E) Average background-subtracted HA intensities of the indicated constructs in D-phase (D) and G1 phase (E). $N = 3$ biological replicates (C–E). Source data are available online for this figure.

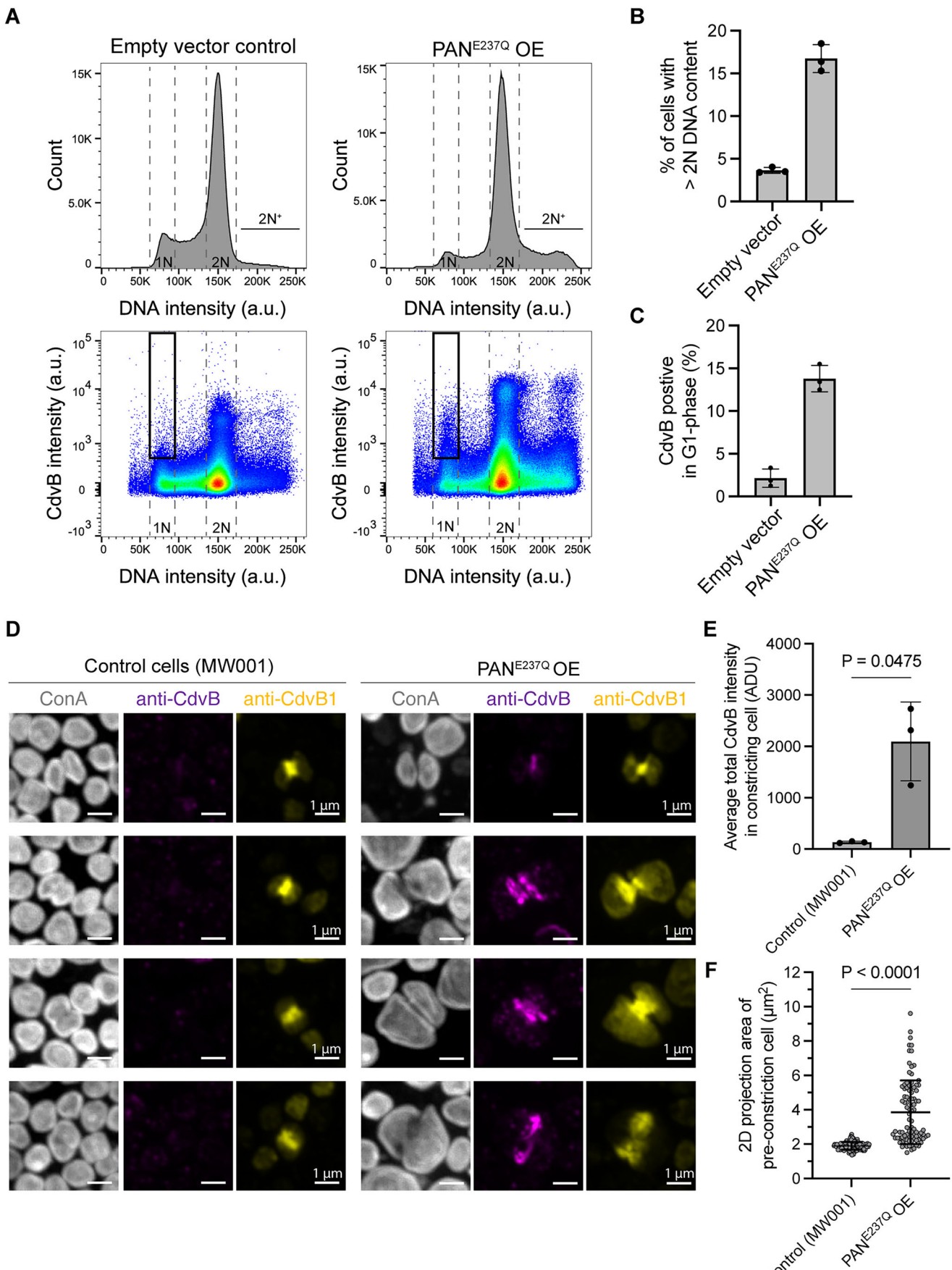

**Fig. 4.  Disruption of PAN activity decreases the degradation of CdvB.**

(**A**) Representative flow cytometry histograms and scatter plots of the dominant negative mutant of PAN (PAN$^{E237Q}$) overexpression after 4 h addition of arabinose; $n = 5.0 \times 10^5$ events each. (**B, C**) Percentage of cells with more than 2N DNA content (**B**) and CdvB-positive cells in G1 phase (**C**) from flow cytometry analysis in (**A**). (**D**) Example immunofluorescence images (maximum projection) of membrane constricting cells from MW001 background strain control (left) and HA-PAN$^{E237Q}$ overexpression (right). Scale bars: 1 μm. (**E**) Quantification of total CdvB intensity of constricting cells in (**D**). Welch $t$ test; $N = 3$ biological replicates. ADU: analog digital unit. (**F**) Quantification of the area of cells (maximum projection) at pre-constriction stage reveals an increase in cell size in HA-PAN$^{E237Q}$ overexpression cells (Mann–Whitney $U$ test, $n = 116$, 94 cells from three biological replicates; $P = 8.9 \times 10^{-26}$). Error bars = mean ± SD. Source data are available online for this figure.

stained and imaged cells expressing the dominant negative PAN$^{E237Q}$ mutant by immunofluorescence microscopy. PAN$^{E237Q}$ expression prevented the degradation of CdvB that typically precedes membrane constriction in *Sulfolobus* cells (Fig. 4D,E). This was accompanied by division failure in many cells, resulting in an increase in cell size (Fig. 4F) and the accumulation of polyploid cells (Fig. 4B). Encouraged by these results implicating PAN in the timely degradation of CdvB, we performed a cell cycle synchronization experiment and Western blotting analysis (Fig. 5A) to determine how PAN levels change during cell cycle progression in *Sulfolobus*. After release from a G2 acetic acid arrest, levels of PAN peaked at ~100 min after release, coinciding with a major population of cells in D-phase (Fig. 5A,B). This suggests the possibility that the oscillation in PAN levels could contribute to the cyclic change in CdvB stability. To disrupt the normal pattern of PAN expression, we tested the effects of overexpressing HA-PAN from an arabinose-inducible promoter. This led to a decrease in the population of cells expressing high levels of CdvB protein, which typically correspond to the cells in the pre-constriction phase of division (Fig. 5C black boxes, Fig. 5D), as expected if CdvB was subject to premature degradation. As expected following the premature loss of CdvB, HA-PAN overexpression also led to an increase in the percentage of cells with >2N DNA content indicative of division failure (Fig. 5E).

Concomitant with this study looking at the mechanism of cyclic CdvB degradation, we discovered a putative ArsR (arsenical resistance operon repressor) family transcription factor, *saci_0800* (NCBI gene ID: 78441147, UniProt accession: Q4JAK9), that when overexpressed from the arabinose promoter induces the accumulation of CdvB protein in G1 cells (Fig. 6A–C)—a phenotype similar to the one caused by the PAN$^{E237Q}$ dominant negative mutant. We named this factor "Cell Cycle Control Transcription Factor 1" (CCTF1). This increase in CdvB levels in G1 phase of cells overexpressing CCTF1 did not result from a transcriptional upregulation since the RT-qPCR analysis showed ~30% reduction of CdvB transcript level when CCTF1 was overexpressed (Fig. 6D). This finding led us to consider whether CCTF1 might instead alter CdvB protein levels by influencing PAN expression. This prediction was borne out by RT-qPCR and Western blot analyses, which showed that CCTF1 overexpression strongly reduces the levels of PAN transcripts and PAN protein (Fig. 6E,F). Consistent with this, CCTF1 overexpression also mimicked the phenotype of expression of the PAN dominant negative (E237Q) by preventing the complete degradation of CdvB during membrane constriction (Fig. 6G,H), leading to an increase in CdvB protein levels even though levels of CdvB transcript were reduced. Importantly, the CCTF1 transcription factor was previously reported to have a cyclic expression pattern (Lundgren and Bernander, 2007), with the peak of expression also in D-phase, similar to its homolog in the related

species *Saccharolobus islandicus* (Gomez-Raya-Vilanova et al, 2025). These results imply that cyclic changes in PAN expression, mediated in part by changes in the levels of the transcription factor CCTF1, may underlie the switch in free CdvB monomer stability that occurs as cells move from a ring assembly stage (where CdvB is relatively stable) to the constriction phase (when CdvB is unstable).

## Discussion

The onset of cytokinesis in *Sulfolobus* is accompanied by rapid proteasomal-mediated degradation of CdvB, a non-contractile ESCRT-III polymer (Tarrason Risa et al, 2020), whose degradation plays a key role in triggering cell division. Here, using an in vivo flow cytometry-based degradation assay to search for sequences in CdvB that are sufficient to confer cyclic degradation on a fusion protein, we have mapped the regulation to the C-terminal region of CdvB. The tail of CdvB is conserved across Sulfolobales and is able to recapitulate the pattern of rapid protein degradation as cells pass from D-phase to G1 phase. Importantly, this part of the protein is also absent from the other *Sulfolobus* ESCRT-III paralogs, which are degraded as cells pass from G1 into S-phase once their transcription has ended. Examining individual truncation constructs revealed two important features of the CdvB C-terminal tail: (i) The C-terminal region of CdvB is sufficient to confer on a protein a high basal rate of degradation across all stages of the cell division cycle, implying the presence of general *cis*-acting degradation signals; and (ii) the presence of the C-terminal broken winged-helix domain of CdvB is associated with stabilization of a fusion protein prior to division.

These observations imply that the broken-winged helix temporarily inhibits CdvB degradation pre-division in a way that contributes to the cyclic accumulation of the protein. While an earlier study showed that disassembly of the CdvB polymer is likely a prerequisite for its proteasome-mediated degradation (Hurtig et al, 2023; Liu et al, 2025), the stabilization due to the broken-winged helix proved to be independent of CdvB polymerization, since fusion proteins that lack the ESCRT-III core domain are not associated with the division ring, but still undergo a switch in stability as they pass from pre-division into G1. Intriguingly, this is the same region of the protein that was previously shown to bind to the E3B motif in CdvA (Samson et al, 2011). This suggests the possibility that, during ring assembly, which is triggered by the accumulation of CdvA in early division phase of the cell cycle, the ability of the E3B motif to complete the winged helix fold may stabilize CdvB to prevent its proteasome-mediated degradation. When we imaged proteins that contain the CdvB C-terminal tail (Fig. 2E) or the broken winged-helix domain fused to LacS (Δ149–193 construct, Appendix Fig. S8), these fusion proteins

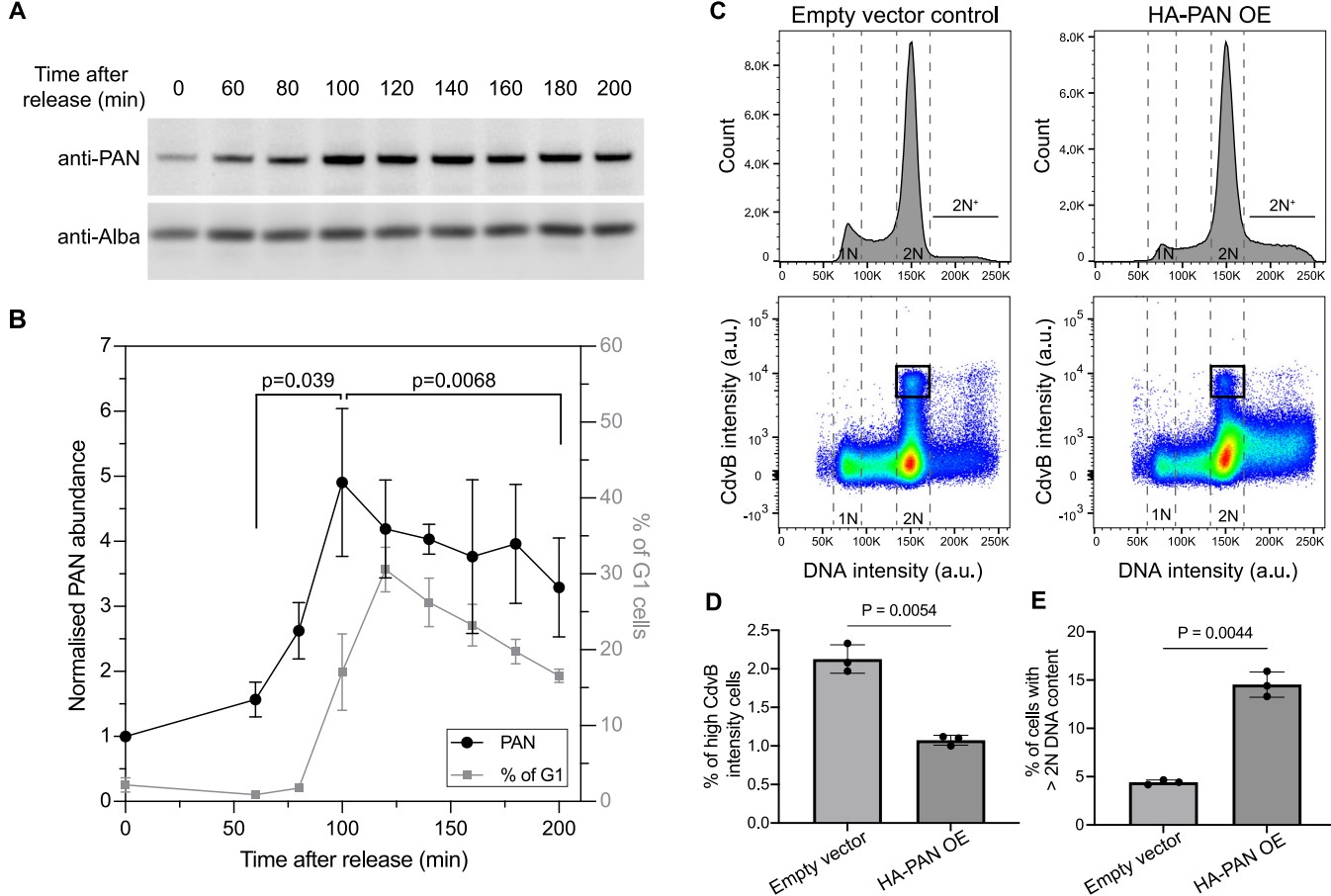

**Fig. 5. PAN expression is subject to cell cycle-dependent regulation.**

(A, B) PAN exhibits cyclic expression in synchronized cells (MW001) after release from acetic acid-induced G2 arrest, as indicated by representative Western blots showing the level of PAN and Alba loading control (A). Quantification of PAN expression level from the western blots is shown in (B), together with a quantification of the percentage of cells in G1 phase shown in gray. Note that PAN levels peak slightly earlier than the peak of G1 cells, indicating it reaches the highest expression at around D-phase when CdvB is degraded. The relative PAN protein level is normalized by loading control (Alba) and time point = 0 min. Ratio paired *t* test, N = 3 biological replicates. (C–E) Representative flow cytometry histograms and scatter plots of HA-PAN overexpression from an arabinose-inducible promoter after 4 h induction. HA-PAN overexpression leads to a decrease in the proportion of cells with high levels of CdvB (black boxes in (C), quantified in (D)), and to an increase in the percentage of cells with a > 2 N DNA content (E). Welch *t* test, N = 3 biological replicates each. Error bars = mean ± SD. Source data are available online for this figure.

appeared diffuse in most cells, with only few cells showing partial colocalization with CdvA (Appendix Fig. S8, yellow arrows). This lack of colocalization could reflect the modest binding affinity of the broken-winged helix for the CdvA E3B motif (Samson et al, 2011) and/or the presence of a soluble pool of CdvA that stabilizes CdvB monomers. However, we cannot exclude the existence of other proteins that interact with the broken winged-helix domain of CdvB to mediate the switch in its stability during division.

Our data also point to additional factors contributing to the change in CdvB degradation rate as cells undergo division. As cells pass from late G2 into G1, there is an increase in the level of PAN — the disassembly factor that unfolds proteins as a prelude to proteasome-mediated degradation. This likely plays an important role in the regulation of cyclic CdvB levels, since our data reveal that PAN is required to ensure that CdvB is completely degraded during division and does not persist until G1. We speculate that these different factors, i.e., degradation signals intrinsic to CdvB, the protection from degradation when present in the polymeric

state and cell cycle changes in the levels of PAN, all contribute to the switch-like changes in the rate of CdvB degradation that occurs as *S. acidocaldarius* cells divide. These rapid changes in CdvB stability are important as they allow the initial buildup of cellular CdvB concentration during the pre-division phase as cells are constructing a template division ring, which serves as a platform for the recruitment of CdvB1 and CdvB2, and the constriction of the CdvB1 and CdvB2 rings following CdvB degradation (Fig. 7A,B).

These data lead us to propose the following model to explain the changes in protein stability that accompany division. As G2 cells enter D-phase there is a wave of Cdv protein expression, which includes CdvA, CdvB, and CdvC (Vps4). As CdvA assembles into a ring, CdvB is stabilized by its broken-winged helix (perhaps via interaction with CdvA and/or other unknown factors). This allows CdvB to form an ESCRT-III polymer (via its ESCRT-III core domain), which further hinders degradation of the monomeric protein by the proteasome, leading to the formation of a robust CdvB ring. Subsequently, as cells pass a regulatory decision point

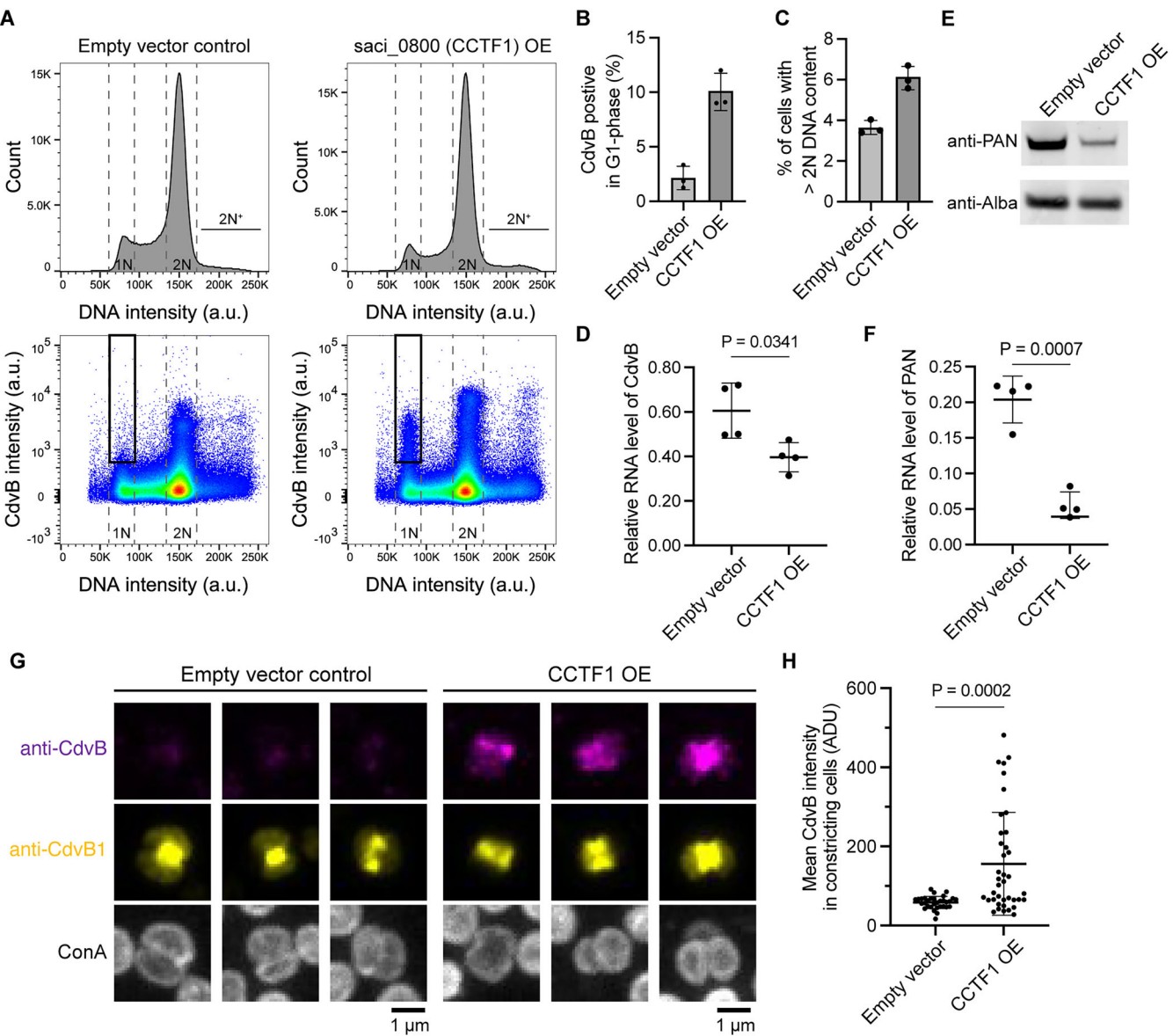

**Fig. 6. Overexpression of CCTF1 suppresses the degradation of CdvB.**

(**A–C**) Example flow cytometry histograms and scatter plots of CCTF1 (*saci_0800*) overexpression (**A**) with quantification of CdvB-positive cells in G1 phase (**B**) and percentage of cells with more than 2N DNA content (**C**). N = 3 biological replicates; error bars: mean ± SD. Note that the empty vector controls here are the same dataset as in Fig. 4A–C and are repeated here for clarity. (**D**) RT-qPCR analysis of CdvB in empty vector and CCTF1 overexpression strains after 4 h addition of arabinose (Welch *t* test, N = 4 biological replicates; error bars: mean ± SD). (**E**) Representative western blots of PAN protein signal in empty vector control cells and following CCTF1 overexpression. (**F**) RT-qPCR analysis of PAN in empty vector and CCTF1 overexpression strains (Welch *t* test, N = 4 biological replicates; error bars: mean ± SD). (**G**) Examples of the immunofluorescence maximum projection images of empty vector control cells and cells following CCTF1 expression caught mid-constriction. (**H**) Quantification of the mean CdvB intensity mid-constriction in immunofluorescent images like those shown in (**G**). Quantification was performed on maximum projections. Mann–Whitney *U* test (n = 32, 39 cells from three biological replicates; error bars: mean ± SD). Source data are available online for this figure.

(Parham et al, 2025), the CdvB polymer is then disassembled by rising levels of Vps4 and separated from CdvA. This generates a pool of free CdvB monomer that can then be unfolded by the PAN AAA-ATPase, whose levels accumulate as a result of the decrease in CCTF1 (Lundgren and Bernander, 2007). In this way, the increase in PAN levels during the late D-phase facilitates the rapid proteasome-mediated degradation of CdvB. The loss of CdvB from the division ring frees CdvB1/B2 polymers to constrict and drive cytokinesis (see Fig. 7A,B for the proposed model). Note that while

downregulation of PAN by CCTF1 overexpression leads to reduced degradation of CdvB, further studies are needed to elucidate precisely how CCTF1 and other cyclic transcription factors work together to regulate sharp transitions in cell cycle states.

In parallel to this work, a recent study using the closely related species *Sa. islandicus* as a model system suggested that cell cycle-dependent phosphorylation of the 20S proteasome subunits can regulate its assembly and interaction with PAN to modulate degradation activity during division (preprint: Huang et al, 2025).

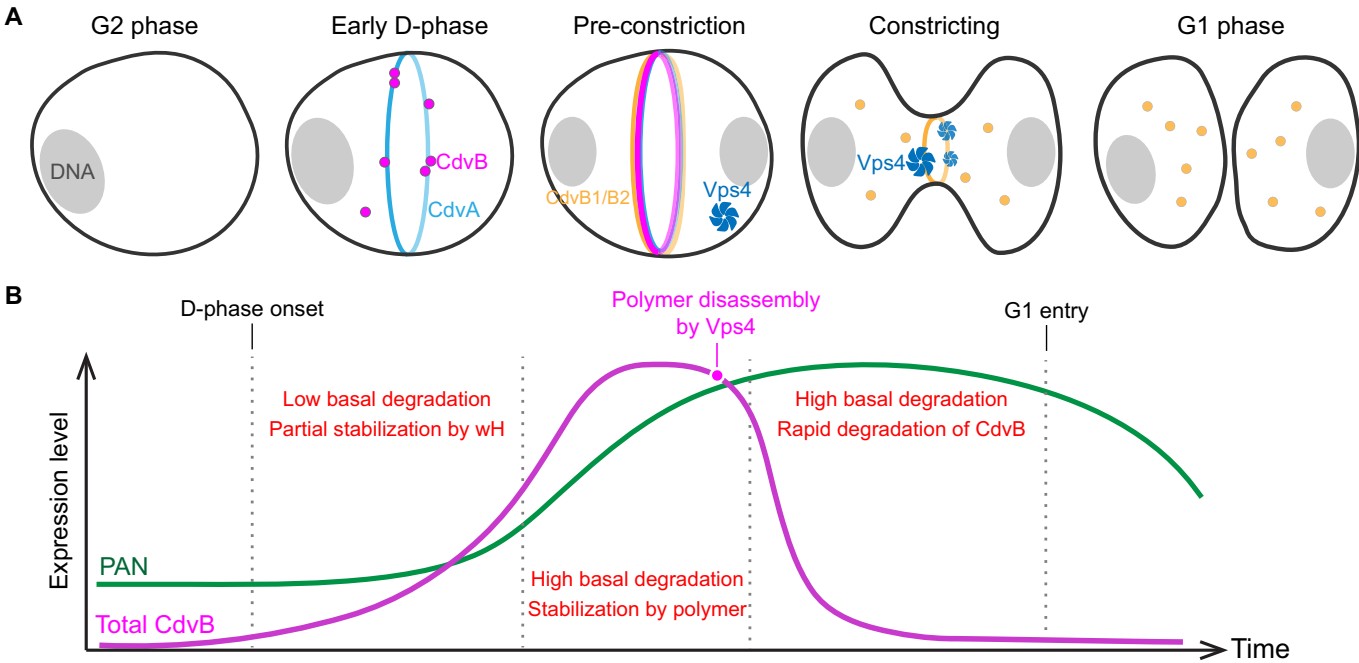

**Fig. 7. Proposed model for cyclic degradation of CdvB.**

(A) Graphic shows the state of the division ring at key cell cycle stages from G2 phase until the completion of cell division. (B) Schematic diagram of the change in CdvB (magenta) and PAN (green) abundance during division.

While these phosphorylation sites are not conserved across Sulfolobales, their effect is expected to mirror the impact of cyclic PAN expression we demonstrate here in *S. acidocaldarius*. Taken together these data suggest that there may be multiple levels of control that all contribute to cyclic protein degradation in these archaea are they pass through division and into G1. In this light, we also note that a residual pool of CdvB was reported to remain in *Sa. islandicus* cells as they divide (Liu et al, 2025), even though CdvB degradation is required for division in this organism just as it is in *S. acidocaldarius*. Thus, there may be subtle differences in the kinetics of CdvB degradation across lineages. Interestingly, the effect of the AAA-ATPase PAN on the degradation of CdvB resembles the role played by the AAA-ATPase-associated protease, ClpXP, in the degradation of the division protein, FtsZ, in *Caulobacter crescentus* (Williams et al, 2014), suggesting that cell cycle-dependent proteolysis may be perform a common regulatory function in all three domains of life. These different studies also highlight the evolutionary plasticity of the proteasome regulatory strategies that can be used to control cell cycle progression.

In conclusion, in this study we demonstrate that archaea can achieve robust cyclic degradation of a key cell division machinery during each round of the cell cycle through a combination of substrate-specific molecular signals, polymer dynamics, and the cyclic expression of regulatory components of the protein degradation machinery. Our results provide a new paradigm for understanding proteasome-mediated cell cycle regulation mechanisms prior to the evolution of CDK/cyclins and E3 ubiquitin ligases. In conjunction with the transcriptional control of cell cycle-dependent gene expression (Lundgren and Bernander, 2007; Li et al, 2023), these molecular mechanisms likely constitute an ordered cell cycle system that is functionally analogous to the case in eukaryotes.

## Methods

### Reagents and tools table

| Reagent/resource | Reference or source | Identifier or catalog number |
|---|---|---|
| **Experimental models** | | |
| *Sulfolobus acidocaldarius* | Wagner et al, 2012 | MW001 |
| **Recombinant DNA** | | |
| pSVAara-FX-STOP | van der Kolk et al, 2020 | N/A |
| pSVAara-FX-HA | van der Kolk et al, 2020 | N/A |
| pSVAara-HA-LacS | This study | N/A |
| pSVAara-HA-LacS-CdvB[C-term] | This study | N/A |
| pSVAara-HA-LacS-CdvB1[C-term] | This study | N/A |
| pSVAara-HA-LacS-CdvB2[C-term] | This study | N/A |
| pSVAara-CdvB-HA | This study | N/A |
| pSVAara-HA-LacS-CdvB[C-term] | This study | N/A |
| pSVAara-HA-LacS-CdvB[C-term Δ183-193] | This study | N/A |
| pSVAara-HA-LacS-CdvB[C-term Δ194-261] | This study | N/A |
| pSVAara-HA-LacS-CdvB[C-term Δ212-253] | This study | N/A |

| Reagent/resource | Reference or source | Identifier or catalog number |
|---|---|---|
| pSVAara-HA-LacS-CdvB$^{C\text{-term }\Delta160\text{-}183}$ | This study | N/A |
| pSVAara-HA-LacS-CdvB$^{C\text{-term }\Delta184\text{-}261}$ | This study | N/A |
| pSVAara-HA-LacS-CdvB$^{C\text{-term }\Delta194\text{-}211}$ | This study | N/A |
| pSVAara-HA-LacS-CdvB$^{C\text{-term }\Delta149\text{-}193}$ | This study | N/A |
| pSVAara-HA-PAN$^{E237Q}$ | This study | N/A |
| pSVAara-HA-PAN | This study | N/A |
| pSVAara-CCTF1-HA | This study | N/A |
| **Antibodies** | | |
| Anti-CdvB serum | Tarrason Risa et al, 2020 | N/A |
| Anti-CdvB1 IgY | Tarrason Risa et al, 2020 | N/A |
| Anti-CdvB2 IgG (peptide antibody) | Tarrason Risa et al, 2020 | N/A |
| Anti-CdvA IgY | Tarrason Risa et al, 2020 | N/A |
| Anti-CdvA serum | This study | N/A |
| Anti-PAN serum | Robinson Lab | N/A |
| Anti-HA Monoclonal Antibody (2-2.2.14) | Invitrogen | 26183 |
| Anti-Alba serum | Tarrason Risa et al, 2020 | N/A |
| Anti-Rabbit IgG, AF488 | Invitrogen | A11034 |
| Anti-Mouse IgG, AF488 | Invitrogen | A11029 |
| Anti-Mouse IgG, AF647 | Invitrogen | A21235 |
| Anti-Chicken IgY, AF546 | Invitrogen | A11040 |
| Anti-Chicken IgY, AF647 | Invitrogen | A21449 |
| Anti-Guinea Pig IgG, AF546 | Invitrogen | A11074 |
| Anti-Mouse IgG IRDye 800CW | LI-COR Biosciences | 926-32210 |
| Anti-Rabbit IgG IRDye 800CW | LI-COR Biosciences | 926-32211 |
| **Oligonucleotides and other sequence-based reagents** | | |
| RT-qPCR primers | Merck | See Appendix Table S2 |
| **Chemicals, enzymes, and other reagents** | | |
| NZ-Amine AS | Sigma-Aldrich | N4517 |
| Bortezomib | Abcam | Ab142123 |
| DMSO | Sigma-Aldrich | D8418 |
| L-Arabinose | Sigma-Aldrich | A3256 |
| Gelzan CM (Gelrite) | Sigma-Aldrich | G1910 |
| NcoI-HF | New England Biolabs | R3193S |
| XhoI | New England Biolabs | R0146S |
| Gibson Assembly Master Mix | New England Biolabs | E2611S |
| Gene Pulser/MicroPulser Electroporation Cuvettes | Bio-Rad | 1652089 |

| Reagent/resource | Reference or source | Identifier or catalog number |
|---|---|---|
| Q5 High-Fidelity DNA Polymerase | New England Biolabs | M0491S |
| Hoechst 33342 | Invitrogen | H3570 |
| DAPI | Invitrogen | D3571 |
| Concanavalin A AF647 | ThermoFisher | C21421 |
| Polyethyleneimine | Sigma-Aldrich | P3143 |
| 4x Laemmli Sample Buffer | Bio-Rad | 1610747 |
| NuPAGE Bis-Tris Mini Protein Gels, 4–12% | Invitrogen | NP0321, NP0322, NP0323 |
| TRIzol | Invitrogen | 15596026 |
| Turbo DNase | Invitrogen | AM2238 |
| RNeasy Mini Kit | Qiagen | 74104 |
| Luna Universal One-Step RT-qPCR Kit | New England Biolabs | E3005S |
| **Software** | | |
| Fiji | Schindelin et al, 2012 | fiji.sc |
| Flowjo v10 | BD Biosciences | www.flowjo.com |
| GraphPad Prism 10 | Dotmatics | www.graphpad.com |
| Unipro UGENE v52.1 | Okonechnikov et al, 2012 | http://ugene.net/ |
| **Other** | | |
| *E. coli* DH5-alpha | New England Biolabs | C2987U |
| *E. coli* ER1821 | Lab collection | N/A |

## Cell culture and growth conditions

All *S. acidocaldarius* strains were grown in Brock medium at 75 °C, pH 3.0 with constant shaking. Growth media is supplemented with 0.1% NZ-amine, 0.2% sucrose, and with sulfuric acid used to adjust the pH to 3 (Brock et al, 1972). MW001, a uracil auxotrophic *S. acidocaldarius* strain (Wagner et al, 2012) used as the background strain in this study was grown with 4 mg/L uracil supplemented. MW001 is an auxotrophic mutant generated from DSM639 strain (NCBI taxonomy ID: 330779).

For all experiments, cells were grown to $OD_{600}$ of 0.1–0.2, corresponding to the exponential growth phase. Cells were treated with 10 μM or 1 μM bortezomib (Abcam catalog number ab142123; stock solution 100 mM in DMSO), for proteasomal inhibition experiments. Induction of protein expression was performed by supplementation with the addition of 0.2% arabinose.

Synchronization of cells was performed by supplementing the media with 2 mM acetic acid for 4.5 h. Cells were released from synchronization by pelleting at 5000 rcf for 4 min at 60 °C. Cell pellets were washed by resuspending in fresh pre-warmed media and pelleting at 5000 rcf for 4 min three times, before being diluted back into fresh, pre-warmed Brock medium.

Cells were fixed on ice by three stepwise additions of ice-cold ethanol to the final concentration of 70% as previously described

(Cezanne et al, 2023). Fixed samples were stored at 4 °C for up to three months.

## Molecular genetics

CdvB protein truncation fragments were procured commercially (gBlocks gene fragment, IDT). All fragments were designed with 30 bases of overlap with the plasmid backbone at both 5' and 3' ends. Fragments were assembled using Gibson Assembly (New England Biolabs, E5510S) in pSVAara-FX-stop plasmid double-digested with NcoI and XhoI. Similarly, cloning of saci_0800-HA over-expression construct was performed by Gibson assembly with custom synthesized gBlock containing codon optimized *saci_0800* fragment flanked with 30 base pairs of overlapping sequencing for insert and double digested (with NcoI and XhoI) pSVAara-FX-HA plasmid as the receiving vector. All cloned plasmids were verified by Sanger sequencing. Plasmids were methylated in vivo by transforming into *E. coli* ER1821. Methylated plasmids were then used for transforming electrocompetent MW001 cells by electroporation (2000 V, 25 µF, 600 ohms, 1 mm). Positive colonies were selected on gelrite-Brock plates without uracil supplementation, followed by validation of PCR genotyping with Sanger sequencing.

## Immunostaining

In total, 1 mL of fixed cells was pelleted at 8000 rcf for 3 min. Cell pellets were resuspended in PBS supplemented with 0.2% Tween-20 and 3% bovine serum albumin (PBSTA) and washed twice. After washing, cells were resuspended in 200 µL of primary antibodies (Appendix Table S1) in PBSTA overnight at 23 °C and 500 rpm. Cells were washed three times with PBST (without bovine serum albumin) and resuspended in 200 µL of secondary antibodies (Appendix Table S1) in PBSTA and/or 50–200 µg/ml Concanavalin A conjugated to Alexa Fluor 647 for cell surface labeling (ThermoFisher, C21421) for 3 h at 23 °C and 500 rpm shaking. Cells were washed three times with PBSTA and resuspended in PBST with 2 µM Hoechst 33342 (Invitrogen Cat. H3570) for visualization of DNA for flow cytometry or 2 µM DAPI (Invitrogen, D3571) for microscopy imaging.

## Flow cytometry

Fixed cells were stained as described above. All flow cytometry analysis was performed on a BD Biosciences LSRFortessa. Laser wavelengths of 355, 488, 561, and 633 nm were used together with filters 450/50 (ultraviolet), 525/50 (blue), 582/15 (yellow-green), and 670/14 (red), respectively. In addition, side scatter and forward scatter signals were recorded. Collected data were analyzed using FlowJo v10.10. Single cells were first selected by excluding the off-diagonal data points in the height vs. area scatter plot in the Hoechst channel. 1 N and 2 N DNA content were assigned by the peaks in the DNA intensity histograms based on Hoechst stain intensity. For HA-tag intensity measurement, the D-phase population was selected by gating the highest CdvA intensity populations with 2 N DNA content which typically correspond to ~5% of total cells, and G1 cells were gated by selecting populations with 1 N DNA content using the Hoechst channel. The average HA intensity of each population was measured by the FlowJo statistics function. Average HA intensities of the MW001 control in the corresponding

gates were then used as staining background and were subtracted to yield the background-subtracted HA signal. G1/D-phase HA intensity ratio was then calculated as the background-subtracted G1-phase HA intensity divided by the background-subtracted D-phase HA intensity.

## Confocal microscopy

Imaging was performed on a Nikon Eclipse Ti2 inverted microscope equipped with a Yokogawa SoRa scanner unit and Prime 95B scientific complementary metal-oxide semiconductor (sCMOS) camera (Photometrics). For this, cells were imaged in Lab-Tek #1 chambered coverglass coated with 1% polyethylenei-mine for 2 h at 37 °C. Images were acquired using 100X oil immersion objective (Apo TIRF 100X/NA = 1.49, Nikon). The 2.8X magnification lens in the SoRa unit allowed a total magnification of 280x. For labeled proteins and DNA labels, cells were exposed to the laser for 50 to 200 ms and 500 ms, respectively. Laser power was adjusted accordingly to prevent saturation of pixels. Data for the *z*-axis were gathered by taking 10 frames with a 0.22-µm step. Image analysis and *z*-axis sum or maximum projections were performed with Fiji.

## Western blotting

Cells were lysed in 1× Laemmli buffer (Bio-Rad), followed by incubation of samples at 99 °C for 10 min and centrifugation to remove cell debris. Samples were then loaded on a NuPAGE 4–12% Bis-Tris gel (Invitrogen) and run at 150 V with MES-SDS running buffer. Transfer of proteins to the nitrocellulose membrane was done by 100 V for 1 h at 4 °C. Membranes were blocked with 5% milk in PBS supplemented with 0.2% Tween-20 (PBST). After blocking, membranes were incubated with primary antibodies (Appendix Table S1) in PBST with 5% milk at 4 °C overnight. The *Sulfolobus* PAN antibody serum was shared by Dr. Nick Robinson (Lancaster University). Membranes were washed three times with PBST for 5 min after which membranes were incubated with 1:10,000 secondary antibodies in PBST with 5% milk at room temperature for 2 h. Signal from proteins was recorded by exposing membranes using the Bio-Rad ChemiDoc system, and the band intensity was analyzed by the gel analysis tool in Fiji.

## RNA extraction and RT-qPCR

For total RNA extraction, the cell pellet from 20 mL of culture was resuspended with 750 µL of TRIzol (Invitrogen), and the subsequent chloroform extraction and RNA precipitation were performed following the manufacturer's protocol. The total RNA pellet was then dissolved in 100 µL of 1× TURBO DNase digestion buffer and treated with TURBO DNase (0.04 U/µL; Invitrogen, #AM2238) for 20 min at 37 °C. The RNA was then purified by using the RNeasy kit (Qiagen) and the purity and concentration determined with a nanodrop before storage in −70 °C.

RT-qPCR reaction was prepared by using Luna One-step RT-qPCR kit (New England Biolab) with 5–10 ng of total RNA per reaction and run on the ViiA 7 Real-Time qPCR system (Applied Biosystems). The relative RNA level was quantified by the $2^{-\Delta\Delta Ct}$ method using SecY (*saci_0574*) as the housekeeping gene. The primer pairs used are summarized in Appendix Table S2.

## Quantification and statistical analyses

All data processing and statistical analysis were performed using Microsoft Excel and GraphPad Prism 10 software. The Welch *t* test was used to incorporate possible differences in standard deviations, and the Mann–Whitney *U* test was used when $n > 6$ without assuming a normal distribution of data. All statistical tests performed were two-sided. Three to four biological replicates were used for all quantitative experiments. All experiments in the main figures were independently repeated at least two times in the laboratory. No blinding was used for data collection.

## Data availability

This study includes no data deposited in external repositories.

The source data of this paper are collected in the following database record: biostudies:S-SCDT-10_1038-S44318-025-00688-7.

## Peer review information

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

## Acknowledgements

All flow cytometry experiments were performed at the Medical Research Council Laboratory of Molecular Biology Flow Cytometry facility, and we would like to thank members of the Flow Cytometry Facility for their technical support. We would also like to thank Dr. Nick Robinson (Lancaster University) for sharing the *Sulfolobus* PAN antibody. JT was supported by a collaborative VolkswagenStiftung "Life?" collaborative Research Grant awarded to BB (94933). Y-WK was supported by an EMBO postdoctoral fellowship (ALTF 903-2021) and by the Medical Research Council - Laboratory of Molecular Biology (MC_UP_1201/27); SF was supported by the Wellcome Trust (222460/Z/21/Z); BB received support for work in *Sulfolobus* from the Medical Research Council - Laboratory of Molecular Biology (MC_UP_1201/27), the Wellcome Trust (222460/Z/21/Z). BB was also supported by a Moore-Simons Project on the Origin of the Eukaryotic Cell, Simons Foundation (735929LPI).

## Author contributions

**Yin-Wei Kuo**: Conceptualization; Formal analysis; Validation; Investigation; Writing—original draft; Writing—review and editing; Directed the project, analyzed data for LacS fusion experiments, proteasome inhibition, PAN mutant, synchronization and overexpression experiments, discovered and characterized CCTF1. **Jovan Traparić**: Conceptualization; Resources; Formal analysis; Validation; Investigation; Generated all LacS fusion strains and carried out fusion-protein analysis together with proteasome inhibition and PAN mutant experiments. **Sherman Foo**: Resources; Writing—original draft; Writing —review and editing; Assisted with the generation of strains and manuscript writing. **Buzz Baum**: Conceptualization; Supervision; Funding acquisition; Writing—original draft; Project direction and administration; Writing—review and editing.

Source data underlying figure panels in this paper may have individual authorship assigned. Where available, figure panel/source data authorship is listed in the following database record: biostudies:S-SCDT-10_1038-S44318-025-00688-7.

## Disclosure and competing interests statement

The authors declare no competing interests.

