## [Peer Review File · The EMBO Journal]

The mechanism of cell-cycle-dependent proteasomal degradation of archaeal ESCRT-III homolog CdvB in *Sulfolobus*

Yin-Wei Kuo, Jovan Traparić, Sherman Foo, and Buzz Baum

Corresponding author(s): Buzz Baum (bbaum@mrc-lmb.cam.ac.uk)

Review Timeline:

Transfer Date:	20th Oct 25
Editorial Decision:	20th Nov 25
Revision Received:	13th Dec 25
Accepted:	18th Dec 25

Editor: Hartmut Vodermaier

Transaction Report: This manuscript was transferred to The EMBO Journal following peer review at Review Commons.

Review
COMMONS

Review #1

1. Evidence, reproducibility and clarity:

Evidence, reproducibility and clarity (Required)

In this manuscript, Kuo and colleagues investigate the molecular basis for CdvB degradation during the cell cycle in the model archaeon *Sulfolobus acidocaldarius*. They find that the sequence responsible for proteolysis lies within the C-terminal domain of CdvB. CdvB1 and B2, that form the contractile ring and are not degraded, are devoid of that domain. Paradoxically, the same C-terminal portion also seems to protect CdvB from degradation prior to the onset of division through a broken winged helix domain. Lastly, they identify the role of a transcription factor, CCTF1, in regulating the expression level of an activator of the proteasome, PAN, thereby controlling CdvB proteolysis.

****Major points:****

- It is unclear to me what is included in the CdvB C-terminal domain in the constructs used here. The alpha-5 domain seems to be either included (see Fig. 3), or excluded (see Fig. 1 and rest of the text). This is critical to the authors' interpretations.

- Throughout the manuscript, the hypothesis about a potential interaction between CdvB and CdvA is confusing and needs to be clarified, as this is a crucial point in the paper's conclusions. Page 5, the authors write that the cytoplasmic localization of HA-LacS-CdvBC-ter is consistent with it lacking the ESCRT-III domain required for polymerization, meaning that the ESCRT-III domain mediates membrane recruitment. This is in contradiction i) with Figure 3 that describes CdvBC-ter as comprising this domain ($\alpha 5$) along with MIM2 and the bwH domains, and ii) with page 4 where it is said that the C-terminal domain consists of MIM2, a linker, and the bwH domain and has been reported to interact with CdvA at the plasma membrane; not that the alpha-5 is required to be recruited. Therefore, it should rather be unexpected that the C-terminal domain of CdvB is found to be cytoplasmic and doesn't interact with CdvA at the plasma membrane (Fig. 2E). Later in the text, page 7, page 8 and in the discussion page 13, the authors postulate that the bwH domain must stabilize CdvB during early division through its ability to bind to CdvA. Again page 9, they describe the bwH domain as able to bind CdvA at very high levels. This is inconsistent with the data presented in Fig. 2E that clearly shows that CdvB C-terminal domain does not in fact seem to interact with CdvA. Further in the discussion page 14 and 15, it is written that CdvB must be stabilized through its ability to incorporate into the ESCRT-III polymer, i.e. not because it interacts with CdvA. Could the authors clarify which domain is expected to mediate membrane recruitment, and through what partner

(CdvA versus other ESCRT-III proteins), and re-interpret their data accordingly? Lastly, instead of comparing the subcellular localization of HA-LacS-CdvBC-ter to that of the full length CdvB, it would be best to test the subcellular localization of a HA-LacS-CdvBFull-length construct, as fusion to the HA-LacS might be interfering with the proper recruitment of CdvBC-ter by CdvA.

- Page 7, the authors write that the bhW domain generally increases the stability of protein constructs in D-phase. This does seem to be the case in Fig. 3C where CdvBC-ter and delta-215-253 show the same level of expression. Could the authors clarify?

- Page 7/ Fig. 3: Fig. 3B shows that the ratio between the expression levels of delta-215-253 in G1 compared to D is close to 0.5, which is interpreted as a lack of cyclic degradation. However, the difference is hardly noticeable in flow cytometry data presented in Fig. 3A where all the dot plots look similar. Maybe it would make it more convincing if the authors presented the expression levels of all the constructs in G1, in a similar way than for phase D in Fig. 3C. In addition, is it at all possible that this specific construct is simply inherently unstable due to folding issues?

- Page 9: When interpreting the phenotype associated with construct delta-149-193, the authors conclude that the bwH domain may interfere with binding of the endogenous CdvB to CdvA, thereby blocking cell division. However, other constructs that possess the bwH domain don't block cell division. Could the authors clarify? Showing the subcellular localization of the endogenous CdvB and/or of the delta-149-193 construct may help interpret this point further.

- Page 9: "We also observed a residual pool of CdvB protein in newly divided G1 cells expressing the dominant negative PAN"; it seems like the levels of CdvB are overall higher in that condition. Could this not suffice to explain this residual pool?

****Minor points:****

- Page 2: "G2 into G1" is confusing. Maybe "progression through the D phase" would be clearer.

- Please define "HA" the first time it is mentioned.

- Pages 3 and 4, the flow cytometry data is overly interpreted as a time series (example "the entire pool of cellular CdvB is degraded before division is complete"). However, cells that lie between G1 and D are either just exiting D or about to enter D. Even if we know the fate of CdvB from previous works, in no case does the flow cytometry data alone supports this chronology.

- How the 1N and 2N populations are determine in the flow cytometry dot plots should be explained. It would make it easier to read if the authors showed in Figure 1 corresponding graphs where cell count is plotted against DNA content. Also, it would be nice that all the

axis ranges be homogenous across all figures.

- The results of the proteasome inhibition experiments are interesting and would be worth including in Figure 1 as opposed to in the supplementary materials. The domain architecture of the CdvB paralogs can be made smaller if space is limited.

- Fig. 2C: It would increase clarity if the authors indicated the percentage of cells in phase D

- Throughout the manuscript: is the G1/D ratio of expression levels normalized to cell count? Please describe.

- Fig. 2E-F: Please label the channels in the microscopy data consistently, using either the staining/antibody used ("ConA", "anti-HA") or the cellular component visualized ("CdvB2", "S-layer"). Please consider labeling each panel with the construct name as well.

- Page 5: replace "fluorescent" by "fluorescence"

- Page 9: "sequences that destabilize the protein"; could the author specify what sequences they are referring to?

- Page 9: "we did not previously observe [...] (Fig. S7A-C)"; this observation is also made in Fig. S1

- Page 10: "These data imply that CdvB accumulates in G1 cells if its degradation is partially compromised"; if this was true, one should expect that a more potent inhibition of protein degradation would lead to an even larger accumulation of CdvB. However, the authors observe that there is no dose-dependent effect at play here. Could the authors propose a different hypothesis?

- Page 12: replace "immunofluorescent" by "immunofluorescence"

- Page 12: "~100 min after release, coinciding with D-phase"; this is misleading. ~100 min after acetic acid release corresponds to the time when a measurable number of cells in the population enter the D-phase.

- Page 12: please indicate the gene accession number for CCTF1; please define ArsR

- Page 12: replace with "Concomitant with the current study, we discovered that a putative ArsR family transcription factor, saci_0800, when over-expressed from the arabinose promoter, induces [...]"

- Page 12: "flat"; do the authors mean stable, constant?

- Page 12: was the expression of PAN also found to be cell-cycle dependent in the Lundgren & Bernander 2007 paper?

- Page 14: The model for the sequence of events in the degradation of CdvB is written in a way that makes it sound like it is a known sequence. Please soften the writing.

- Page 14: Please replace "Vsp4" by "Vps4"

- Figure 6: It would be nice to clearly label the limits of the D-phase. Also, shouldn't Vps4 be represented already present in the cell in the pre-constriction phase? Or maybe include the actual stage where CdvB is degraded.

- Page 15: "in the closely related"
- Page 15: replace "S. islandicus" by "Sa. islandicus"
- Page 15: could the authors briefly describe what is known about proteolysis-dependent control of the cell cycle in bacteria?
- Fig. S4C: was a statistical test performed here? Please show resulting P values.

2. Significance:

Significance (Required)

In this manuscript, Kuo and colleagues reveal novel molecular mechanisms involved in the regulation of the archaeal cell cycle. It was previously shown that in the model Thermoproteota *Sulfolobus acidocaldarius*, progression through the division phase is controlled by the proteasome-dependent degradation of specific target proteins, including CdvB, a component of the ESCRT-III-based cytokinetic ring. The details of how proteins are targeted to the proteasome are lacking. Overall, the work is well-conducted and reports novel findings that are of great interest in the field. However, I believe some major points should be addressed to make the main conclusions clear.

3. How much time do you estimate the authors will need to complete the suggested revisions:

Estimated time to Complete Revisions (Required)

(Decision Recommendation)

Between 1 and 3 months

Yes

Review #2

1. Evidence, reproducibility and clarity:

Evidence, reproducibility and clarity (Required)

I have no concerns about the reproducibility of the data.

The manuscript is clear and well written.

I have some questions about how data was analyzed in Figures 2 and 3 related to fold changes in the abundance of CdvB truncation mutants.

Summary- this manuscript is focused on the cyclical degradation of the escrt-III homolog CdvB protein in the archaeal relative of eukaryotes, *Sulfolobus acidocaldarius*. This degradation has been established to be important for cytokinesis and abscission. The authors map the domain on CdvB that is responsible for its degradation, and show that the expression of PAN, which is similar to the cap on eukaryotic proteasome, is both itself cyclically expressed, and also important for CdvB degradation.

****Major Points:****

1. The authors state that CdvB1 and B2 are degraded minutes after CdvB, as cells pass G1/S. But this is hard to say based purely on the protein level measurements in Figure 1, done at a single time point. I would suggest restating so as to describe their data, but saying that this is consistent with prior experiments showing timing of destruction.
2. The data in Figure 3A is difficult to interpret. The HA signal, based on the intensity graphs, is clearly low for all of the deletion mutants tested. However, the difference in the 2n/1n intensity appears no different between them. It is unclear how this is being calculated. Specifically, the last four intensity plots appear similar with respect to the signal difference between the 2N and 1N regions, and it is unclear how the delta 212-255 could be 5x different.
 - a. A similar question is related to Figure 2B- it is unclear how these comparisons are being done. The CdvB2 cterm appear to have a similar decrease relative to CdvB based on the intensity plots.
3. The intensity measurements in Fig 3 are also difficult to interpret because so many of the cells are at zero on the Y-axis. I realize this might have been done to be able to compare the LacS control, but it means comparisons between the CdvB-cterm and the relevant mutants, is difficult to assess because the signal is severely compressed.
4. Data showing how the winged helix domain specifically can increase fusion protein stability in 3C is unclear. Could that not be said of all of the mutants?

5. Is the data in Fig 3D directly comparable to the other truncations tested in this figure (was this flow analyzed at the same time under identical conditions)? If so, why aren't they all shown together and quantified in the graphs in 3B and 3C?

6. It is possible that the effects of CCTF1 and PAN on CdvB are less direct than what is suggested. While this is a matter of preference, it might be interesting to discuss different possibilities, highlighting both direct and indirect impacts of these factors on the stability of CdvB.

****Minor Points:****

1. The schematic depiction of the domains with relevant amino acids, which is shown in Fig 3E, would be very helpful to have earlier, even just as panel A in this figure.

2. Significance:

Significance (Required)

- This work begins to spell out the requirements for degradation of CdvB.
- In addition, the authors show the proteasome cap PAN is cell cycle regulated, is influenced by the transcription factor CCTF1, and that this regulated CdvB. Although it is unclear how direct these connections might be.
- I have expertise in cell cycle control in eukaryotes and the role of ubiquitin mediated protein degradation.

3. How much time do you estimate the authors will need to complete the suggested revisions:

Estimated time to Complete Revisions (Required)

(Decision Recommendation)

Less than 1 month

4. Review Commons values the work of reviewers and encourages them to get credit for their work. Select 'Yes' below to register your reviewing activity at Web of Science Reviewer Recognition Service (formerly Publons); note that the content of your review will not be visible on Web of Science.

No

Review #3

1. Evidence, reproducibility and clarity:

Evidence, reproducibility and clarity (Required)

****Summary:****

The authors report the demonstration that the C terminal broken winged helix domain of CdvB, an ESCRT-III homolog used in cell division in the model archaeon *Sulfolobus acidocaldarius*, is sufficient to destabilize the protein, leading to rapid proteolysis of the CdvB protein before cell division. The broken winged helix domain of CdvB was previously shown to interact with the E3B helix of CdvA, a non ESCRT-homolog that is the first to mark the midcell pre-division and which recruits CdvB to form a ring at the midcell in early division. The authors also describe a novel cyclically expressed transcription factor which they term "CCTF1" (*saci_0800*) to act as a repressor to the expression of the proteasome-activating nucleotidase (PAN) (high expression of the PAN has been previously shown to correlate to accelerated degradation of CdvB). This is an excellent and insightful paper. I found it very exciting.

****Major comments:****

1. It would help if the authors would clarify statements about CdvA impacting the stability of CdvB through the broken winged helix domain. In the last paragraph of introduction this sentence is confusing I think because it is very long and says "selectively stabilizing" which is hard to interpret to me at least.

This identifies a portion of the C-terminus of CdvB that renders the protein unstable across the cell cycle and identifies the broken winged helix domain of the protein, which was previously shown to bind CdvA, as playing a role in selectively stabilizing the protein during the pre-division phase of division ring assembly.

In the discussion the authors mention twice this speculative conclusion:

"This suggests the possibility that, during ring assembly, which is triggered by the accumulation of CdvA in early division phase of the cell cycle, the ability of the E3B helix to complete the winged helix fold may stabilize CdvB to prevent its proteasome-mediated degradation."

"We speculate that these different factors including degradation signals intrinsic to CdvB, its protection from degradation when present in a polymer, [...] all contribute to the switch-like change in the rate of CdvB degradation at division in *S. acidocaldarius*."

This study may benefit from showing evidence to support the speculation that the completion of the broken winged helix by the E3B helix of CdvA stabilizes CdvB during ring assembly such as a protein fragment complementation assay.

The claims are otherwise convincingly supported by the data and experiments shown. The methods are reported in a clear and concise manner and are feasibly reproducible.

2. I was quite confused by the cdvB intensity distribution showing one high peak of intensity above 10^4 and the majority in 10^3 in Figure 1 but nowhere else. What is the interpretation of this peak. It is not mentioned that I saw.

3. Figure 1 A,B,&C should include either in the flow cytometric legend or in the figure description a clarification of what method of fluorescent label was used; for example, immunostained protein under its native promoter versus an HA tagged protein under an induced promoter on a plasmid. This may be important for interpretation of these graphs against the graphs later shown. Similarly In figure 2 E and F label the construct not just the tag.

****Minor comments:****

1. Missing a citation on page 2 line 16 "the broken winged helix domain of the protein, which was previously shown to bind CdvA"

2. "The loss of CdvB then allows the CdvB1/B2 cytokinetic ring to constrict." Be sure that "allows" is the intended verb here. It implies an inherent contractile elasticity in the CdvB1/B2 cytokinetic ring and that CdvB acts as a force that holds the CdvB1/B2 ring open when it is assembled. It may be clearer to the readers if the authors make their intended interpretations of the pre-existing models clear.

3. Figure 6B could benefit from the addition of indicators for cell cycle stage to make the proposed model more straightforward for readers to interpret.

4. In two places authors write, "identifies the broken winged helix domain of the protein, which was previously shown to bind CdvA, as playing a role in selectively stabilizing the protein during the pre-division phase of division ring assembly." and "we identify the terminal broken winged helix of CdvB, which was previously shown to bind CdvA, as a domain that is sufficient to render a fusion protein unstable as cells transit from division phase to G1 phase." The wording of this major point in each instance could stand to be edited. The use of the words "identifies/identify" could temporarily mislead a reader into a false notion that the broken winged helix domain itself was identified by the authors in this paper, when this paper is more accurately describing the determination that the broken winged helix, which has been previously characterized and known to crucially interact with CdvA, is also a destabilizing factor to CdvB itself.

2. Significance:

Significance (Required)

This paper is a dignified stepwise advancement in the current body of knowledge surrounding the steps involved in the closely controlled cell division cycle of a model archaeon. The two major claims the authors make are useful insights into the basic physiology of the organism; that the broken winged helix domain of CdvB contributes to the protein's self-degradation signaling, and the preliminary characterization of a transcription factor that represses the nucleotidase implicated in the degradation of CdvB.

The audience that this paper is most suitable for are researchers interested in basic cell biological research in the archaeal domain and fundamental physiological science. This research could be used by others in the field to design experiments that target specific points of progression in the cell cycle of the model archaeon *Sulfolobus acidocaldarius*. It is a useful, practically applicable, and advancement in the field's understanding of the regulation of the cell cycle in this organism.

3. How much time do you estimate the authors will need to complete the suggested revisions:

Estimated time to Complete Revisions (Required)

(Decision Recommendation)

Less than 1 month

No

Full Revision

Manuscript number: RC-2025-03148

Corresponding author(s): Buzz Baum

[Please use this template only if the submitted manuscript should be considered by the affiliate journal as a full revision in response to the points raised by the reviewers.]

*If you wish to submit a preliminary revision with a revision plan, please use our "Revision Plan" template. **It is important to use the appropriate template to clearly inform the editors of your intentions.**]*

1. General Statements [optional]

This section is optional. Insert here any general statements you wish to make about the goal of the study or about the reviews.

We were pleased to see that the Reviewers all understood the main aims of our paper and appreciated the importance of determining the mechanism of cell cycle dependent proteasome-mediated protein degradation. We also appreciated their constructive criticism. To address the concerns raised, our revised paper now includes the following new experimental data:

Figure 1E and F: Flow cytometry scatter plots of DMSO control and bortezomib treated MW001 cells have been moved into the main Figure (formerly Figure S1).

Figure S1: Representative time-lapse flow cytometry analysis of synchronized MW001.

Figure 3E: HA intensities of truncated constructs in G1 phase.

Figure 5D-F: New experiments showing that PAN overexpression reduces levels of CdvB.

Figure S8: Representative immunofluorescence images of HA-LacS-CdvBC-term Δ 149-193.

In response to the shared feedback from Reviewers, we have also reorganized the way the data presented in Fig. 3 and Fig. S5 and have rewritten the associated section to improve the clarity of this complex but important part of the paper. Finally, a new paragraph has been added to the Discussion to explain the likely role of interactions between CdvA and the CdvB broken winged helix for the cyclic regulation of CdvB levels.

This section is mandatory. Please insert a point-by-point reply describing the revisions that were already carried out and included in the transferred manuscript.

Reviewer #1 (Evidence, reproducibility and clarity (Required)):

In this manuscript, Kuo and colleagues investigate the molecular basis for CdvB degradation during the cell cycle in the model archaeon *Sulfolobus acidocaldarius*. They find that the sequence

responsible for proteolysis lies within the C-terminal domain of CdvB. CdvB1 and B2, that form the contractile ring and are not degraded, are devoid of that domain. Paradoxically, the same C-terminal portion also seems to protect CdvB from degradation prior to the onset of division through a broken winged helix domain. Lastly, they identify the role of a transcription factor, CCTF1, in regulating the expression level of an activator of the proteasome, PAN, thereby controlling CdvB proteolysis.

Major points:

- It is unclear to me what is included in the CdvB C-terminal domain in the constructs used here. The alpha-5 domain seems to be either included (see Fig. 3), or excluded (see Fig. 1 and rest of the text). This is critical to the authors' interpretations.

The boundaries of the reported MIM2 motif (Samson *et al.* 2008) and $\alpha 5$ helix from the predicted (AlphaFold AF-Q4J924-F1) and recent cryo-EM structure (Drobnič *et al.* biorxiv 2025) show partial overlap. To avoid truncating the MIM2 motif due to deletion of $\alpha 5$ helix, we included this region in LacS C-terminal fusion proteins despite $\alpha 5$ helix is part of the conserved ESCRT-III core domain. We now clarify our construct design rationale in the text (page 5, line 18).

- Throughout the manuscript, the hypothesis about a potential interaction between CdvB and CdvA is confusing and needs to be clarified, as this is a crucial point in the paper's conclusions.

We reorganized our data associated with Fig. 3 and Fig. S5, and rewrote the corresponding section of the text to better explain the results of the LacS fusion experiments.

Page 5, the authors write that the cytoplasmic localization of HA-LacS-CdvBC-ter is consistent with it lacking the ESCRT-III domain required for polymerization, meaning that the ESCRT-III domain mediates membrane recruitment. This is in contradiction i) with Figure 3 that describes CdvBC-ter as comprising this domain ($\alpha 5$) along with MIM2 and the bwH domains,

In the CdvB^{C-term} construct, only $\alpha 5$ helix of the ESCRT-III core domain was included, which based on structure should not be sufficient to induce its polymerization (Drobnič *et al.* biorxiv 2025). Thus, we do not expect LacS-CdvB^{C-term} to colocalise with CdvB/B1/B2.

and ii) with page 4 where it is said that the C-terminal domain consists of MIM2, a linker, and the bwH domain and has been reported to interact with CdvA at the plasma membrane; not that the alpha-5 is required to be recruited. Therefore, it should rather be unexpected that the C-terminal domain of CdvB is found to be cytoplasmic and doesn't interact with CdvA at the plasma membrane (Fig. 2E). Later in the text, page 7, page 8 and in the discussion page 13, the authors postulate that the bwH domain must stabilize CdvB during early division through its ability to bind to CdvA. Again page 9, they describe the bwH domain as able to bind CdvA at very high levels. This is inconsistent with the data presented in Fig. 2E that clearly shows that CdvB C-terminal domain does not in fact seem to interact with CdvA.

The lack of colocalization of LacS-CdvB^{C-term} and the CdvA is likely due to the rather modest affinity of the CdvA E3B motif for the CdvB wH domain (K_d of $\sim 6 \mu\text{M}$ or lower affinity depends on the construct used, Samson *et al.* 2011). This is compounded by the fact that the expression levels of LacS-CdvB^{C-term} are low, due to the low intrinsic stability of this sequence, we expect limited colocalization to the division ring.

By contrast, the markedly elevated expression level of LacS-CdvB^{C-term Δ 149-193} likely increases the binding between CdvA and wH domain of LacS-CdvB^{C-term Δ 149-193} (also see response below).

While these results can be most easily explained by the wH domain of CdvB providing partial stabilization during D-phase through its previously identified interaction with CdvA, as stated in the text (and made clear in the second paragraph of the Discussion) it is also possible that other factors contribute to this process.

Further in the discussion page 14 and 15, it is written that CdvB must be stabilized through its ability to incorporate into the ESCRT-III polymer, i.e. not because it interacts with CdvA. Could the authors clarify which domain is expected to mediate membrane recruitment, and through what partner (CdvA versus other ESCRT-III proteins), and re-interpret their data accordingly?

To clarify: the interaction between CdvB and CdvA was first studied by Samson et al., 2011. CdvB is recruited to the medial division ring by CdvA. The data presented here suggests that the ability of CdvA to do this is likely mediated, in part, by the ability of CdvA to stabilize the protein against degradation. At the same time, the ability of CdvB to form an ESCRT-III polymer, also renders the protein stable as was shown previously by Hurtig et al 2023. Thus, it is likely that the interaction of CdvA with the broken winged helix of CdvB, the impact of this on CdvB stability, and CdvB capacity for polymerisation all combine to enable the construction of a CdvB ring.

Lastly, instead of comparing the subcellular localization of HA-LacS-CdvBC-ter to that of the full length CdvB, it would be best to test the subcellular localization of a HA-LacS-CdvBFull-length construct, as fusion to the HA-LacS might be interfering with the proper recruitment of CdvBC-ter by CdvA.

Since full length CdvB forms a polymer that is resistant to protein degradation (Hurtig et al. 2022), the purpose of the LacS fusion assay is to separate the contribution of polymer formation from the intrinsic effect of C-terminal sequence. Additionally, the strong division defects observed in HA-LacS-CdvB^{C-termΔ149-193} construct (Fig. S5) mirror the effects of CdvAΔE3B overexpression (Parham et al. 2025), making it clear that LacS does not abolish interactions between wH domain and CdvA.

- Page 7, the authors write that the bhW domain generally increases the stability of protein constructs in D-phase. This does seem to be the case in Fig. 3C where CdvBC-ter and delta-215-253 show the same level of expression. Could the authors clarify?

We have modified the text and data to make the logic clear. In brief, our flow cytometry data show that there is ~20% D-phase signal decrease in Δ215-253 compared to the full length CdvB C-term in Fig. 3D. This is also reflected in the histogram in Fig. S6, where the D-phase HA signal distribution shows a slight leftward shift for the Δ215-253 construct. We have also added the relevant control data to show the HA intensity of CdvB^{C-term} in Fig. 3D and 3E to aid a direct visual comparison. Additionally, the HA-Δ183-193 construct is expressed at higher levels in D-phase, while having a similar low level expression in the G1 phase, consistent with the proposed stabilization effect of the broken winged helix in D-phase. All of these data support a role for the broken winged helix in protein stabilization in D-phase.

- Page 7/Fig. 3: Fig. 3B shows that the ratio between the expression levels of delta-215-253 in G1 compared to D is close to 0.5, which is interpreted as a lack of cyclic degradation. However, the difference is hardly noticeable in flow cytometry data presented in Fig. 3A where all the dot plots look similar. Maybe it would make it more convincing if the authors presented the expression levels of all the constructs in G1, in a similar way than for phase D in Fig. 3C.

Full Revision

We agree. This is a good idea. We have now included the plot of HA intensity in G1 phase as Fig. 3E as suggested. This makes it clear that the $\Delta 215-253$ construct is not destabilized upon passage from D into G1.

In addition, is it at all possible that this specific construct is simply inherently unstable due to folding issues?

In G1 phase, levels of HA- $\Delta 215-253$ are slightly higher than those of full length HA-CdvB^{C-term} (Fig. 3E). This argues against the idea that this construct is inherently unstable.

- Page 9: When interpreting the phenotype associated with construct delta-149-193, the authors conclude that the bwH domain may interfere with binding of the endogenous CdvB to CdvA, thereby blocking cell division. However, other constructs that possess the bwH domain don't block cell division. Could the authors clarify? Showing the subcellular localization of the endogenous CdvB and/or of the delta-149-193 construct may help interpret this point further.

LacS-CdvB^{C-term $\Delta 149-193$} is present at very high levels in D-phase, consistent with a model in which the presence of wH provides stronger stabilization in D-phase. As can be seen from Fig. 3D and S5C, these levels are much higher than other construct containing the wH (~4 times of full C-terminal tail, and ~2 times of $\Delta 183-193$). We think is mainly due to loss of the MIM2-linker region, which acts to enhance the instability of the protein across the cycle as shown by the HA- $\Delta 215-253$ construct. It is only when present at very high levels that we think it perturbs division.

We speculate that this is due to the fact that the affinity between the CdvB wH domain and CdvA E3B motif is rather modest (also see response above). This is supported by the fact that immunofluorescence imaging shows that the majority of cells have a diffuse pattern of HA- $\Delta 149-193$ molecules (HA signal in Fig. S8), while in a few cells we also observed partial colocalization of HA signal with high intensity CdvA foci (Fig. S8, yellow arrows). Thus, we propose it is only in cells expressing $\Delta 149-193$ construct that the broken winged helix is present at sufficient levels to interfere with CdvA-B interactions and trigger the checkpoint (Parham et al., 2025). We now include a discussion of clarification in the main text (page 10, line 8-10).

- Page 9: "We also observed a residual pool of CdvB protein in newly divided G1 cells expressing the dominant negative PAN"; it seems like the levels of CdvB are overall higher in that condition. Could this not suffice to explain this residual pool?

As seen from Fig. 5A, PAN is expressed throughout the cell cycle with peak expression at D-phase. We thus expect dominant negative PAN to affect the stability in all phases, including D and G1 phase (like bortezomib treatment). As observed in Fig. 4A, the CdvB levels in PAN dominant negative mutant are higher than empty vector in D-phase, consistent with the decrease in PAN activity. Given that the increase in CdvB level in D-phase is an expected consequence of reduced PAN activity, the question whether the residual pool of CdvB in G1 phase originates from the reduced PAN activity or increase CdvB level becomes circular. To further test the role of PAN we also overexpressed HA-tagged PAN from the arabinose-inducible promoter (new Fig. 5D-F). The overexpression of PAN decreased cells with high CdvB intensity and division defect, according with increase in CdvB degradation. Combining both loss-of-function experiments of PAN by dominant-negative mutant and CCTF1 overexpression, and gain-of-function experiment by HA-PAN overexpression, we think our result strongly support the role of PAN in the degradation of CdvB at the D/G1 phase transition.

Minor points:

Full Revision

- Page 2: "G2 into G1" is confusing. Maybe "progression through the D phase" would be clearer.

Corrected as suggested.

- Please define "HA" the first time it is mentioned.

Added.

- Pages 3 and 4, the flow cytometry data is overly interpreted as a time series (example "the entire pool of cellular CdvB is degraded before division is complete"). However, cells that lie between G1 and D are either just exiting D or about to enter D. Even if we know the fate of CdvB from previous works, in no case does the flow cytometry data alone supports this chronology.

As shown in the example time course analysis of synchronised cells in Fig. S1, the flow cytometry plots provide a way to distinguish cells in the early and late division phase. Early division phase is characterized by cells with positive but lower CdvB and CdvB1/B2 signal with 2N DNA content, while late division cells can be subdivided into pre-constriction stage (high CdvB and high CdvB1/B2 with 2N DNA) and constriction stage (low CdvB and high CdvB1 with 2N DNA). The synchronization time course matches with the asynchronized cells pattern, and we think our statement regarding CdvB degradation timing is supported by our flow cytometry analysis.

- How the 1N and 2N populations are determine in the flow cytometry dot plots should be explained. It would make it easier to read if the authors showed in Figure 1 corresponding graphs where cell count is plotted against DNA content. Also, it would be nice that all the axis ranges be homogenous across all figures.

To make this clearer, we now include a more detailed description in the method section and the DNA histogram in Fig. 1 as suggested. We are unable to use the same axis range throughout all figures since different experiments used different fluorophore-conjugated secondary antibodies and different voltage settings on the flow cytometry detection. For experiments conducted together with direct comparison, we used the same axis range for the same staining channel.

- The results of the proteasome inhibition experiments are interesting and would be worth including in Figure 1 as opposed to in the supplementary materials. The domain architecture of the CdvB paralogs can be made smaller if space is limited.

We thank the reviewer for this suggestion. We have now moved the proteasome inhibition result to the main figure (Fig. 1E,F) as suggested.

- Fig. 2C: It would increase clarity if the authors indicated the percentage of cells in phase D

We agree. We have now included the percentage of cells in the figure legend as suggested.

- Throughout the manuscript: is the G1/D ratio of expression levels normalized to cell count? Please describe.

For HA-tag intensity measurement the D-phase population was selected by gating the highest CdvA intensity populations with 2N DNA content which typically correspond to ~5% of total cells, and G1 cells were gated by selecting populations with 1N DNA content using the Hoechst channel. The average HA intensity of each population was measured by FlowJo statistic function. Average HA intensities of the MW001 control in the corresponding gates were then used as staining background and were subtracted to yield the background subtracted HA signal. G1/D-phase HA intensity ratio was then calculated as the background subtracted G1 phase HA intensity

Full Revision

divided by the background subtracted D-phase HA intensity. We have now included this detailed description of our analysis in the method section for clarity.

- Fig. 2E-F: Please label the channels in the microscopy data consistently, using either the staining/antibody used ("ConA", "anti-HA") or the cellular component visualized ("CdvB2", "S-layer"). Please consider labeling each panel with the construct name as well.

We have changed our labelling as suggested.

- Page 5: replace "fluorescent" by "fluorescence"

Done.

- Page 9: "sequences that destabilize the protein"; could the author specify what sequences they are referring to?

We have added " [...] regulated by sequences within the MIM2 and linker region that render the protein unstable independent of cell cycle stage, and by the broken winged helix domain that preferentially stabilizes the protein during the ring assembly phase of division" to be more specific.

- Page 9: "we did not previously observe [...] (Fig. S7A-C)"; this observation is also made in Fig. S1

The reference for the figure has been added.

- Page 10: "These data imply that CdvB accumulates in G1 cells if its degradation is partially compromised"; if this was true, one should expect that a more potent inhibition of protein degradation would lead to an even larger accumulation of CdvB. However, the authors observe that there is no dose-dependent effect at play here. Could the authors propose a different hypothesis?

High dosage of proteasome inhibitor leads to a strong division arrest at the pre-constriction phase (Tarrsaon-Risa *et al.* 2020) and thus reduces the number of cells entering G1. This explains why G1 cells with residual pool of CdvB were primarily observed at low dose of proteasome inhibitor. Under these conditions cells are still able to complete cytokinesis in the presence of residual CdvB. By contrast, strong inhibition of CdvB degradation locks the cells primarily in the division phase prior to cytokinesis. To make this clear we have added a clarification to the figure legend (Fig. S7) to explain the superficial lack of dose-dependency.

- Page 12: replace "immunofluorescent" by "immunofluorescence"

Changed.

- Page 12: "~100 min after release, coinciding with D-phase"; this is misleading. ~100 min after acetic acid release corresponds to the time when a measurable number of cells in the population enter the D-phase.

For clarity we have changed it to "[..], coinciding with a major population of cells in D-phase".

- Page 12: please indicate the gene accession number for CCTF1; please define ArsR

The accession number and full name of ArsR have been added as suggested.

Full Revision

- Page 12: replace with "Concomitant with the current study, we discovered that a putative ArsR family transcription factor, *saci_0800*, when over-expressed from the arabinose promoter, induces [...]"

Changed.

- Page 12: "flat"; do the authors mean stable, constant?

Correct. We have rephrased the sentence to avoid confusion.

- Page 12: was the expression of PAN also found to be cell-cycle dependent in the Lundgren & Bernander 2007 paper?

PAN was not included in the selected genes for microarray analysis in Lundgren & Bernander's study.

- Page 14: The model for the sequence of events in the degradation of CdvB is written in a way that makes it sound like it is a known sequence. Please soften the writing.

We have now adjusted the sentence to specify the time sequence of events is a proposed model.

- Page 14: Please replace "Vsp4" by "Vps4"

Thanks. Corrected.

- Figure 6: It would be nice to clearly label the limits of the D-phase. Also, shouldn't Vps4 be represented already present in the cell in the pre-constriction phase? Or maybe include the actual stage where CdvB is degraded.

We have now marked the onset and exit of D-phase as suggested in our model scheme (Fig. 7).

- Page 15: "in the closely related"

Changed.

- Page 15: replace "*S. islandicus*" by "*Sa. islandicus*"

Changed.

- Page 15: could the authors briefly describe what is known about proteolysis-dependent control of the cell cycle in bacteria?

Thanks for the suggestion. We have now briefly discussed the analogy between the *C. crescentus* cell cycle-dependent degradation of bacterial division protein FtsZ via AAA ATPase ClpX and the degradation of CdvB facilitated by PAN in the discussion as suggested (page 18, line 9-13).

- Fig. S4C: was a statistical test performed here? Please show resulting P values.

Included.

Reviewer #1 (Significance (Required)):

In this manuscript, Kuo and colleagues reveal novel molecular mechanisms involved in the regulation of the archaeal cell cycle. It was previously shown that in the model Thermoproteota *Sulfolobus acidocaldarius*, progression through the division phase is controlled by the

Full Revision

proteasome-dependent degradation of specific target proteins, including CdvB, a component of the ESCRT-III-based cytokinetic ring. The details of how proteins are targeted to the proteasome are lacking. Overall, the work is well-conducted and reports novel findings that are of great interest in the field.

We thank the Reviewer for their assessment.

However, I believe some major points should be addressed to make the main conclusions clear.

We have now done this.

Reviewer #2 (Evidence, reproducibility and clarity (Required)):

I have no concerns about the reproducibility of the data.

The manuscript is clear and well written.

I have some questions about how data was analyzed in Figures 2 and 3 related to fold changes in the abundance of CdvB truncation mutants.

We agree that this was hard to read as presented. We have now modified the text and Figures for this part to make the data and logic clear.

Summary- this manuscript is focused on the cyclical degradation of the escrt-III homolog CdvB protein in the archaeal relative of eukaryotes, *Sulfolobus acidocaldarius*. This degradation has been established to be important for cytokinesis and abscission. The authors map the domain on CdvB that is responsible for its degradation, and show that the expression of PAN, which is similar to the cap on eukaryotic proteasome, is both itself cyclically expressed, and also important for CdvB degradation.

This is a good description of the study.

Major Points:

1- The authors state that CdvB1 and B2 are degraded minutes after CdvB, as cells pass G1/S. But this is hard to say based purely on the protein level measurements in Figure 1, done at a single time point. I would suggest restating so as to describe their data, but saying that this is consistent with prior experiments showing timing of destruction.

We appreciate the comment and now include example of synchronization experiment in Fig. S1, showing that CdvB degradation happens earlier than CdvB1 and the overall degradation timing follow our interpretation from the asynchronized cells shown in Fig. 1. We think this provide more direct support for interpreting the degradation timing of ESCRT-III homologs in the flow cytometry analysis.

2- The data in Figure 3A is difficult to interpret. The HA signal, based on the intensity graphs, is clearly low for all of the deletion mutants tested. However, the difference in the 2n/1n intensity appears no different between them. It is unclear how this is being calculated. Specifically, the last four intensity plots appear similar with respect to the signal difference between the 2N and 1N regions, and it is unclear how the delta 212-255 could be 5x different.

We agree that the data were hard to interpret as presented. To make this easier, we have reorganized our data presentation and writing in Fig. 3 and Fig. S5 to improve the clarity.

Additionally, we added the intensity of HA in G1 phase in Fig. 3E, and added reference lines of CdvB^{C-term} HA intensity in Fig. 3D and 3E for easier comparison. As can be seen from the average HA intensity, $\Delta 212-253$ showed slight decrease in HA intensity in D-phase (~20%), while show ~50% increase in G1-phase (Fig. 3D, 3E). This thus gives rise to an increase in G1/D-phase ratio shown in Fig. 3C. We have also expanded the method section to have more detailed description of our analysis method (see response to Reviewer 1 above).

a. A similar question is related to Figure 2B- it is unclear how these comparisons are being done. The CdvB2 cterm appear to have a similar decrease relative to CdvB based on the intensity plots.

The G1/D-phase HA intensity ratio is based on measurement of average HA intensity in D-phase (gated by CdvA intensity) and G1 phase (gated by DNA staining intensity). The detail analysis steps have been included in the method section.

3- The intensity measurements in Fig 3 are also difficult to interpret because so many of the cells are at zero on the Y-axis. I realize this might have been done to be able to compare the LacS control, but it means comparisons between the CdvB-cterm and the relevant mutants, is difficult to assess because the signal is severely compressed.

Yes, the scatter plots in Fig. 3B were compressed to aid the direct comparison with the HA-LacS control. To make clear the result, we have now included the measurements of the HA intensity for both D-phase and G1 phase (background subtracted) in Fig. 3D and 3E. Additionally, histograms of the HA intensity have been included in Fig. S6 for visualization of the intensity distribution.

4- Data showing how the winged helix domain specifically can increase fusion protein stability in 3C is unclear. Could that not be said of all of the mutants?

We have reorganised the data for clarity. Two pieces of data strongly support this statement. First, the data showing a decrease in HA signal (~20%) in D-phase when wH domain is deleted (HA-LacS-CdvB^{C-term} $\Delta 212-253$) suggest that the wH domain stabilizes CdvB during this phase of the cycle (Fig. 3D). Additionally, consistent with wH domain providing a partial stabilization effect at D-phase, removal of destabilizing sequence at the MIM2 motif from the CdvB^{C-term} (HA-LacS-CdvB^{C-term} $\Delta 149-193$), leads to a large increase in the HA signal (Fig. 3D, 3F). We have rewritten the section and data presentation in Fig. 3 to make this clearer.

5- Is the data in Fig 3D directly comparable to the other truncations tested in this figure (was this flow analyzed at the same time under identical conditions)? If so, why aren't they all shown together and quantified into the graphs in 3B and 3C?

The experiments were performed together. We have now added the quantification of the $\Delta 149-193$ construct to Fig. S5C. Note that we did not include the quantification of HA intensity in G1 phase and the G1/D-phase HA ratio for this experiment because the construct blocks cells from entering G1 population. The G1 population that remains express low levels of $\Delta 149-193$ construct and therefore does not provide a comparable reflection of the stability of this construct at this stage.

6- It is possible that the effects of CCTF1 and PAN on CdvB are less direct than what is suggested. While this is a matter of preference, it might be interesting to discuss different possibilities, highlighting both direct and indirect impacts of these factors on the stability of CdvB.

We agree with this question. To firm up the link between PAN and CdvB, we have performed an additional experiment in which we increase PAN activity by overexpressing HA-PAN from arabinose-inducible promoter. This reduces the percentage of cells with high CdvB intensity (Fig. 5), consistent with our loss-of-function experiments with dominant negative mutant of PAN and overexpression of CCTF1. This also fits with our understanding of PAN in *Sulfolobus* and other systems as a protein that unfolds proteins targeted to the proteasome. Thus, we think the data provide strong support that PAN activity playing a direct role in the degradation of CdvB during cell cycle. The same may not be true for CCTF1 however. Its role is likely to be more indirect, since it is one of a large set of cyclic TFs that appear to regulate transcription as cells divide. Consistent with its proposed effect on PAN, its overexpression reduces levels of PAN protein. However, as a TF with many additional targets it is likely to have complex effects on cells. For this reason, while the loss of function analysis indicates a role for the TF in CdvB degradation, as expected based on the overexpression analysis, we have not included it in this paper because we cannot yet be certain if this is mediated through PAN. We have now commented on this point in the discussion (page 18, line 22-25).

Minor Points:

1- The schematic depiction of the domains with relevant amino acids, which is shown in Fig 3E, would be very helpful to have earlier, even just as panel A in this figure.

Changed as suggested.

Reviewer #2 (Significance (Required)):

- This work begins to spell out the requirements for degradation of CdvB.
- In addition, the authors show the proteasome cap PAN is cell cycle regulated, is influenced by the transcription factor CCTF1, and that this regulated CdvB. Although it is unclear how direct these connections might be.
- I have expertise in cell cycle control in eukaryotes and the role of ubiquitin mediated protein degradation.

This is a good description of our work. We agree that CCTF1 is likely to have more complex effects on division, only part of which are mediated via PAN on CdvB. We have made this clear in the revised manuscript.

Reviewer #3 (Evidence, reproducibility and clarity (Required)):

Summary:

The authors report the demonstration that the C terminal broken winged helix domain of CdvB, an ESCRT-III homolog used in cell division in the model archaeon *Sulfolobus acidocaldarius*, is sufficient to destabilize the protein, leading to rapid proteolysis of the CdvB protein before cell division. The broken winged helix domain of CdvB was previously shown to interact with the E3B helix of CdvA, a non ESCRT-homolog that is the first to mark the midcell pre-division and which recruits CdvB to form a ring at the midcell in early division. The authors also describe a novel cyclically expressed transcription factor which they term ' "CCTF1" (saci_0800)' to act as a repressor to the expression of the proteasome-activating nucleotidase (PAN) (high expression of

Full Revision

the PAN has been previously shown to correlate to accelerated degradation of CdvB). This is an excellent and insightful paper. I found it very exciting.

This is a good summary of our work, and we thank the Reviewer for their assessment.

Major comments:

1. It would help if the authors would clarify statements about CdvA impacting the stability of CdvB through the broken winged helix domain. In the last paragraph of introduction this sentence is confusing I think because it is very long and says "selectively stabilizing" which is hard to interpret to me at least.

This identifies a portion of the C-terminus of CdvB that renders the protein unstable across the cell cycle and identifies the broken winged helix domain of the protein, which was previously shown to bind CdvA, as playing a role in selectively stabilizing the protein during the pre-division phase of division ring assembly.

In the discussion the authors mention twice this speculative conclusion:

"This suggests the possibility that, during ring assembly, which is triggered by the accumulation of CdvA in early division phase of the cell cycle, the ability of the E3B helix to complete the winged helix fold may stabilize CdvB to prevent its proteasome-mediated degradation."

"We speculate that these different factors including degradation signals intrinsic to CdvB, its protection from degradation when present in a polymer, [...] all contribute to the switch-like change in the rate of CdvB degradation at division in *S. acidocaldarius*."

The idea that CdvA regulates the degradation of CdvB via the wH domain is supported by previous biochemical work by Samson et al. 2011 which identified this region as the primary interaction site of CdvB and CdvA. In line with this, cells that overexpress very high levels of the wH arrest in division (this study) as do cells that express a CdvA that lacks the E3B region (Parham et al. 2025). However, the Reviewer is correct that other interpretations are possible. Since based on the data presented we cannot rule out another mechanism that could be CdvA-independent, and limited colocalization was observed between CdvA and LacS fusion constructs containing the CdvB broken winged helix domain (Fig. 2E, Fig. S8; also see response to reviewer 1 and 2 above), we have discussed alternative hypotheses in the discussion section of the revised manuscript to make the speculative nature of this model clear to the audience (second paragraph in Discussion).

This study may benefit from showing evidence to support the speculation that the completion of the broken winged helix by the E3B helix of CdvA stabilizes CdvB during ring assembly such as a protein fragment complementation assay.

The claims are otherwise convincingly supported by the data and experiments shown. The methods are reported in a clear and concise manner and are feasibly reproducible.

This is a good idea. To test this, we performed a complementation assay by fusing the E3B motif of CdvA to the HA-LacS-CdvB^{C-term} construct via a flexible linker and assessed the expression level at D-phase (gated by CdvA and DNA signal as described) after 4 hr induction of arabinose. As shown in the figure below, while this leads to a general increase in the HA intensity of HA-LacS-CdvB^{C-term}-E3B, as expected under a model in which this would protect the broken winged helix from degradation, the difference is not statistically significant ($p=0.15$, $N=3$ replicates). The G1/D-phase HA signal ratios are only marginally different (0.21 vs. 0.27). Since this could also be due to the presence of E3B motif alone, we have chosen not to include the data in the paper.

2. I was quite confused by the *cdvB* intensity distribution showing one high peak of intensity above 10⁴ and the majority in 10³ in Figure 1 but nowhere else. What is the interpretation of this peak. It is not mentioned that I saw.

We apologise if this wasn't made clear. This population corresponds to the pre-constriction phase of division (Tarrason-Risa et al. 2020). We have now added this information to the figure legend.

3. Figure 1 A,B,&C should include either in the flow cytometric legend or in the figure description a clarification of what method of fluorescent label was used; for example, immunostained protein under its native promoter versus an HA tagged protein under an induced promoter on a plasmid. This may be important for interpretation of these graphs against the graphs later shown. Similarly In figure 2 E and F label the construct not just the tag.

Thank you for this suggestion. We have now added the information to figure legends for clarity.

Minor comments:

1. Missing a citation on page 2 line 16 "the broken winged helix domain of the protein, which was previously shown to bind CdvA"

Added.

2. "The loss of CdvB then allows the CdvB1/B2 cytokinetic ring to constrict." Be sure that "allows" is the intended verb here. It implies an inherent contractile elasticity in the CdvB1/B2 cytokinetic ring and that CdvB acts as a force that holds the CdvB1/B2 ring open when it is assembled. It may be clearer to the readers if the authors make their intended interpretations of the pre-existing models clear.

We have changed the wording to "[...] frees the CdvB1/B2 polymers to constrict" to de-emphasise on the putative mechanical effect of CdvB filament.

3. Figure 6B could benefit from the addition of indicators for cell cycle stage to make the proposed model more straightforward for readers to interpret.

We have now added the indicators for the onset and end of D-phase to the figure.

4. In two places authors write, "identifies the broken winged helix domain of the protein, which was previously shown to bind CdvA, as playing a role in selectively stabilizing the protein during

Full Revision

the pre-division phase of division ring assembly." and "we identify the terminal broken winged helix of CdvB, which was previously shown to bind CdvA, as a domain that is sufficient to render a fusion protein unstable as cells transit from division phase to G1 phase." The wording of this major point in each instance could stand to be edited. The use of the words "identifies/identify" could temporarily mislead a reader into a false notion that the broken winged helix domain itself was identified by the authors in this paper, when this paper is more accurately describing the determination that the broken winged helix, which has been previously characterized and known to crucially interact with CdvA, is also a destabilizing factor to CdvB itself.

We have changed the wording to [...] demonstrate [...] to avoid the confusion.

Reviewer #3 (Significance (Required)):

Significance

This paper is a dignified stepwise advancement in the current body of knowledge surrounding the steps involved in the closely controlled cell division cycle of a model archaeon. The two major claims the authors make are useful insights into the basic physiology of the organism; that the broken winged helix domain of CdvB contributes to the protein's self-degradation signaling, and the preliminary characterization of a transcription factor that represses the nucleotidase implicated in the degradation of CdvB.

The audience that this paper is most suitable for are researchers interested in basic cell biological research in the archaeal domain and fundamental physiological science. This research could be used by others in the field to design experiments that target specific points of progression in the cell cycle of the model archaeon *Sulfolobus acidocaldarius*. It is a useful, practically applicable, and advancement in the field's understanding of the regulation of the cell cycle in this organism.

We thank the Reviewer for their positive assessment.

Prof. Buzz Baum
MRC Laboratory for Molecular Biology
Division of Cell Biology
Cambridge Biomedical Campus, Francis Crick Ave, Trumpington
Cambridge, Cambridgeshire CB2 0QH
United Kingdom

20th Nov 2025

Re: EMBOJ-2025-122799-T
The mechanism of cell cycle dependent proteasome-mediated CdvB degradation in *Sulfolobus*

Dear Buzz,

Thank you for submitting your revised Review Commons manuscript for consideration by The EMBO Journal. Given the interest of the subject and the constructive nature of the transferred referee reports, I decided to treat the work like a regular EMBO Journal revision, and returned it directly to the original referees 1 and 2. I am happy to say that both consider the original concerns adequately addressed and no concrete experimental issues remaining. Following adjustment to our specific journal format and incorporation of a few other editorial modifications (as follows), we should therefore be ready to proceed with acceptance and publication of the study:

GENERAL:

- Please download and complete our author checklist (link provided below).
- Please provide suggestions for a short 'blurb' text prefacing and summing up the conceptual aspect of the study in two sentences (max. 250 characters), followed by 3-5 one-sentence 'bullet points' with brief factual statements of key results of the paper; they will form the basis of an editor-written 'Synopsis' accompanying the online version of the article. Please also upload a synopsis image, which can be used as a "visual title" for the synopsis section of your paper. The image should be in PNG or JPG format, and please make sure that it remains in the modest dimensions of (exactly) 550 pixels wide and 300-600 pixels high.
- You shall also receive a separate message from our Source Data curation team, with instructions on how to prepare and upload relevant image and numerical raw data.

TEXT:

- Please upload the manuscript text as an editable DOCX file, without figures included.
- Please adjust the order as well as the headers of the different manuscript sections: Title page with complete author information, Abstract, Keywords, Introduction, Results, Discussion, Methods, Data Availability, Acknowledgements, Disclosure and Competing Interests Statement, References, Main Figure Legends, Tables, Expanded Figure Legends.
- On the abstract page of the manuscript, please include 4-5 general keyword terms to enhance searchability.
- Please double-check to make sure that each figure panel (e.g. Fig. 1A, Fig. 1B) is called out in the text at least once. If referring to a multi-panel figure as a whole, please simply reference "Fig. 7A-B"
- Please note that Materials and Methods need to be described in the main text using our 'Structured Methods' format (for detail, see <https://www.embopress.org/page/journal/14693178/authorguide#structuredmethods>). The in-text "Methods" section should contain method and protocol descriptions (ideally using a step-by-step protocol format to facilitate adoption of the methodologies across labs), while all key reagents, experimental models, software and relevant equipment - including their sources and relevant identifiers - should be listed in a separately uploaded Reagents and Tools Table, a template for which can be downloaded from the above section of our Author Guidelines.
- As we are switching from a free-text author contribution statement towards a more formal statement based on Contributor Role Taxonomy (CRediT) terms, please remove the present Author Contribution section and instead specify each author's contribution(s) directly in the Author Information page of our submission system during upload of the final manuscript. See <https://casrai.org/credit/> for more information.
- Please rename the Competing Interest section into "Disclosure and Competing Interests Statement", in accordance with our

updated Guide to Authors (<https://www.embopress.org/competing-interests>)

- Please also adjust the format for citation of preprints as specified in our author guidelines:

The citation in the text should be: "(preprint: NAME1 et al, YEAR)"

The citation in the reference list: "NAME1, NAME2, ... (YEAR) ARTICLE TITLE. bioRxiv/medRxiv/ResearchSquare(...) doi: XXX"

- Please include a dedicated "Data Availability" section at the end of the Material and Methods (suggested wording: "The [structural coordinates | microarray | mass spectrometry] data from this publication have been deposited to the [name of the database] database [URL] and assigned the identifier [accession | permalink | hashtag]."); should there no data deposition to public repositories linked to the study, this should still be stated as "This study includes no data deposited in external repositories."

DATA:

- Please upload all main Figures as individual, image-only files with sufficient resolution/quality for production.

- Please refer to our author guide (www.embopress.org/page/journal/14602075/authorguide#expandedview) regarding "supplementary information". The current "Supplementary information" PDF should be renamed into "Appendix", and headed by a title page stating "Appendix for [ms title]" and including a table of contents with the page numbers listing the included figures and tables with page numbers. Please make sure to adhere to the nomenclature "Appendix Figure S1/2/3..." and "Appendix Table S1/2" throughout manuscript text and Appendix PDF, and remove/change any mention of "Supplemental/Supplementary" from the Appendix.

- Finally, during routine pre-acceptance checks, our data editors have raised the following queries regarding figures, data, and legends; I would appreciate if you briefly answered to them in the cover letter of your final submission, and made the requested text modifications with changes/additions highlighted via the "Track changes" option, to facilitate our final checking:

1. Please note that the exact p values are not provided in the legend of figure 4F
2. Please indicate the statistical test used for data analysis in the legend of figure 6D
3. Please note that information related to n is missing in the legends of figures 3C-E; 6B, C, D
4. Please note that the error bars are not defined in the legends of figures 6B, C, D

Should you need additional guidance/feedback regarding this final adjustments, please do not hesitate to contact us directly. Thank you again for the opportunity to consider this work for The EMBO Journal, and I look forward to receiving your final version!

With kind regards,

Hartmut

3) Revised manuscript text (including main tables, and figure legends for main and EV figures) has to be submitted as editable

text file (e.g., .docx format). We encourage highlighting of changes (e.g., via text color) for the referees' reference.

4) Each main and each Expanded View (EV) figure should be uploaded as individual production-quality files (preferably in .eps, .tif, .jpg formats). For suggestions on figure preparation/layout, please refer to our Figure Preparation Guidelines:

8) Please note that supplementary information at EMBO Press has been superseded by the 'Expanded View' for inclusion of additional figures, tables, movies or datasets; with up to five EV Figures being typeset and directly accessible in the HTML version of the article. For details and guidance, please refer to:

embopress.org/page/journal/14602075/authorguide#expandedview

9) To facilitate reproducibility and cross-laboratory adoption of methodologies, please structure the Materials & Methods section as outlined in our guide to authors, including a completed Reagents and Tools Table that can be downloaded from our author guidelines as well (<https://www.embopress.org/page/journal/14602075/authorguide#structuredmethods>).

10) Digital image enhancement is acceptable practice, as long as it accurately represents the original data and conforms to community standards. If a figure has been subjected to significant electronic manipulation, this must be clearly noted in the figure legend and/or the 'Materials and Methods' section. The editors reserve the right to request original versions of figures and the original images that were used to assemble the figure. Finally, we generally encourage uploading of numerical as well as gel/blot image source data; for details see: embopress.org/page/journal/14602075/authorguide#sourcedata

Further information is available in our Guide For Authors:

In the interest of ensuring the conceptual advance provided by the work, we recommend submitting a revision within 3 months (18th Feb 2026). Please discuss the revision progress ahead of this time with the editor if you require more time to complete the revisions. Use the link below to submit your revision:

Link Not Available

Referee #1:

The revised manuscript shows substantial improvement, and the authors have satisfactorily addressed my previous concerns. The findings reported in the manuscript by Wu and colleagues are novel and of great importance in the field. I therefore support publication of the current version.

Referee #2:

- the authors have taken steps to largely address most of my technical concerns.

- this represents a very technical description of the features that controls the downregulation of a single protein. This analysis is well done, but might be of limited interest since the specific enzymes involved in the direct regulation is unknown.

Rev_Com_number: RC-2025-03148
New_manu_number: EMBOJ-2025-122799-T
Corr_author: Baum
Title: The mechanism of cell cycle dependent proteasome-mediated CdvB degradation in Sulfolobus

All minor editorial requests have been addressed by the authors.

Prof. Buzz Baum
MRC Laboratory for Molecular Biology
Division of Cell Biology
Cambridge Biomedical Campus, Francis Crick Ave, Trumpington
Cambridge, Cambridgeshire CB2 0QH
United Kingdom

18th Dec 2025

Re: EMBOJ-2025-122799R
The mechanism of cell-cycle-dependent proteasomal degradation of archaeal ESCRT-III homolog CdvB in Sulfolobus

Dear Buzz,

Thank you for submitting your final revised manuscript for our consideration. I am happy to inform you that we have now accepted it for publication in The EMBO Journal.

Please note that I slightly changed the title (see above), solely to make it a bit more explicit and accessible also to non-expert readers.

You may qualify for financial assistance for your publication charges - either via a Springer Nature fully open access agreement or an EMBO initiative. Check your eligibility: <https://link.springer.com/journal/44318/how-to-publish-with-us>

With kind regards,

Hartmut

Please note that it is The EMBO Journal policy for the transcript of the editorial process (containing referee reports and your response letters) to be published as an online supplement to each paper. If you should prefer removal of any referee-only figures included in the point-by-point response(s), e.g. because they may still be used for future publication or because they have been reproduced from published work by others, please do let us know immediately via response email.

More information is available here: <https://link.springer.com/partners/embo-press/editorial-policies#Peer%20review>

Rev_Com_number: RC-2025-03148

New_manu_number: EMBOJ-2025-122799R

Corr_author: Baum

Title: The mechanism of cell-cycle-dependent proteasomal degradation of archaeal ESCRT-III homolog CdvB in Sulfolobus